

# Sea Level Rise in Europe: impacts and consequences

Coordinating lead authors: Roderik van de Wal[1,6] (RvdW), Angélique Melet[2] (AM)

Authors: Debora Bellafiore[3] (DB), Michalis Vousdoukas[4] (MV), Paula Camus[5] (PC), Christian Ferrarin[3] (CF), Gualbert Oude Essink[6,7] (GOE), Ivan D. Haigh[8] (IH), Piero Lionello[9] (PL), Arjen Luijendijk[7] (AL), Alexandra Toimil[10] (AT), Joanna Staneva[11] (JS)

[1] Institute for Marine and Atmospheric Research Utrecht, Utrecht University, Utrecht, the Netherlands
[2] Mercator Ocean International, Toulouse, 31400, France
[3] CNR - National Research Council of Italy, ISMAR – Institute of Marine Sciences, Venice, 30122, Italy
[4] Department of Marine Sciences, University the Aegean, Mitilene, 81100, Greece
[5] Geomatics and Ocean Engineering Group, Department of Sciences and Technologies of Water and Environment, University of Cantabria, 39005, Santander, Spain
[6] Department of Physical Geography, Utrecht University, Princetonlaan 8a, 3584 CB Utrecht, The Netherlands
[7] Deltares, Boussinesqweg 1, 2629 HV Delft, The Netherlands
[8] School of Ocean and Earth Science, University of Southampton, European Way, Southampton, SO14 3ZH, UK
[9] Department of Biological and Environmental Sciences and Technologies, University of Salento, Lecce, 73100, Italy
[10] IHCantabria – Instituto de Hidráulica Ambiental de la Universidad de Cantabria, Isabel Torres 15, 39011, Santander, Spain
[11] Department on Hydrodynamics and Data Assimilation, Helmholtz Zentrum Hereon, GMBH, Geesthacht, 21502, Germany

*Correspondence to*: Roderik van de Wal (R.S.W.vandeWal@uu.nl), Angélique Melet (amelet@mercator-ocean.fr)

**Abstract.**

Sea level rise has major impacts in Europe which vary from place to place and in time depending on the source of the impacts. Flooding, erosion and saltwater intrusion lead via different pathways to various consequences in coastal regions across Europe. Flooding leads via overflow, overtopping and breaching to damage to assets, environment and people. Erosion leads via cliff failure along a different pathway also to damage and saltwater intrusion affects ecosystems and surface waters and salinizes

existing fresh water resources diminishing fresh water availability causing salt damage to crops and health issues to people. This paper provides an overview of the various impacts in Europe.





# 1 Introduction

Sea-level rise (SLR) is a major threat for coastal zones, inducing hazards such as coastal flooding (mild and chronic at high-tides or intense and episodic during storms), permanent submersion of coastal zones, coastal erosion, salt intrusion in surface

and ground water (with adverse impacts on drinkable water and agriculture), problems with water management, and coastal ecosystems degradation or loss (affecting coastal wetlands contributing to coastal zone protection, biodiversity conservation and carbon storage). As sea level is committed to rise over the next centuries (Melet et al., in prep.), coastal zones and communities are expected to be increasingly threatened by sea level changes at various timescales, ranging from episodic extreme events to interannual-to-centennial changes and trends linked to climate change and modes of variability (Bednar-

Friedl et al., 2022; Glavovic et al., 2022; Hallegatte et al., 2013; Le Cozannet et al., 2022; Oppenheimer et al., 2019). Locally, these threats are reinforced by subsidence caused by human activities like groundwater pumping or groundwater fluid extraction for energy purposes (Carbognin & Tosi, 2002).

Impacts of SLR result from the combination of sea-level changes, exposure, and vulnerability (Cardona et al., 2012). Sea level changes with a focus on Europe are discussed in Sea Level Rise in Europe: observations and projections. The increased hazards

posed by SLR in response to climate change have been identified as the main driver of future rise in coastal flood losses, with the relative importance of trends in exposure (related to coastward migration, urbanization and rising asset values) diminishing over time (Vousdoukas et al., 2018b). However, population and economic activities are expected to change as well, in Europe and worldwide possibly also increasing the exposure. Coastal zones are increasingly more densely populated than the hinterland (Small & Nicholls, 2003), exhibit higher rates of population growth and urbanization, which are concentrating

economic assets and critical infrastructures. Coastal migration is driven by the combinations of specific economic, geographic and historical conditions, and includes the concentration of densely settled agricultural areas in well-watered, fertile deltas and coastal plains (Hugo, 2011; McGranahan et al., 2007). The exposure of people and assets to SLR hazards is therefore widespread and increasing; and can thereby extend to a change in the vulnerability to SLR (e.g., urbanization changes the imperviousness of flood-prone areas, Andreadis et al., 2022). Coastal zones in Europe are highly urbanized; more than 30

million people live in the 100-year event flood coastal plain and 50 million in the contiguous and hydrologically connected zone of land along the coast and below 10 m of elevation (sometimes referred to as low-elevation coastal zone (LECZ), (Neumann et al., 2015a) (Figure 1), whether the number of people exposed changes over time depends on assumption on future developments of fertility, mortality and migration.



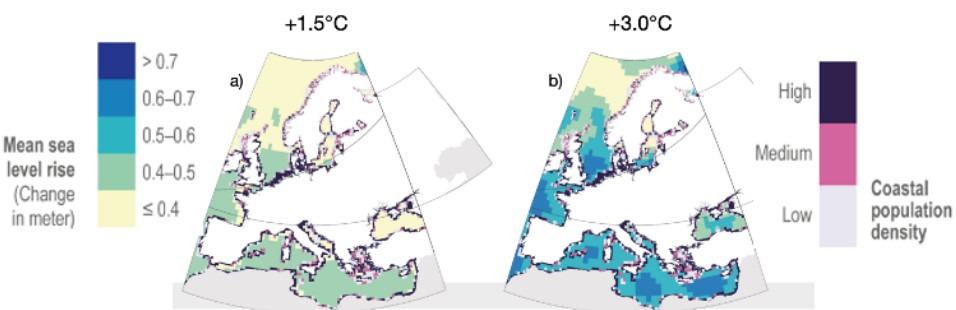

**Figure 1: Mean sea level rise (SLR, in m) and coastal population density in Europe for global warming levels of +1.5∘C and +3∘C. SLR data consider the long-term period (2081-2100) and SSP1-2.6 for (a) and SSP3-7.0 for (b). Adapted from Fig 13.4 of Chapter 13 of AR6 WG2 (Bednar-Friedl et al., 2022).**

Land use planning strategy in coastal lowlands can reduce exposure of EU population to SLR (see Sea Level Rise in Europe: adaptation measures and decision-making principles). For instance, the establishment of coastal setback zones, which are buffer spaces defined by a specific distance from the shoreline's highest water mark where new developments in potentially exposed coastal regions are restricted. Sanò et al. (2011) can reduce the exposure of new urban development by at least 50% in most EU countries by 2100 (Wolff et al., 2023).

Due to the large economic value of coastal zones, economic losses due to coastal flood risks are huge (Abadie et al., 2020; Hallegatte et al., 2013). Presently, the expected annual damage from coastal flooding for Europe alone is around €1.25 billion euros, but could increase by 2-3 orders of magnitudes if coastal adaptation is only maintained to its current level (Vousdoukas et al., 2018a). The impacts of floods (both marine and riverine/ pluvial) on people, the built environment and the economy is one of the four key climate change induced risks identified for Europe (Bednar-Friedl et al., 2022). Deltas are particularly

vulnerable to SLR. In Europe, the main deltas are those of the Rhine-Meuse-Escault (NL), Rhone (FR), Pô (IT), and Ebro (SP) rivers. Rotterdam and London are amongst the world's most exposed cities in terms of population living in the 100-year event flood plain, if there was no flood protection (Hallegatte et al., 2013). In addition to human fatalities, economic losses due to coastal flood risks are considerable. European cities where the annual average losses due to coastal flooding by 2050 will increase particularly assuming present-day defense standards or flood probability, tend to be concentrated along the

Mediterranean coast (Hallegatte et al., 2013). These cities were built close to the shore since historical sea level variability has been low (e.g., small tidal range and interannual variability), as a consequence changes in the mean are felt earlier than in regions where the sea level variability is higher.

Therefore, SLR creates risks for people, ecosystems, land uses, the built environment, and human activities (Le Cozannet et al., 2022). In this paper, a summary of the 6th cycle of the IPCC reports is provided (Section 2). Then, various impacts of SLR

are discussed with a European perspective, using the Source, Pathway, Receptor and Consequence (SPRC) framework



introduced in Section 3. SLR impacts are discussed for coastal flooding (Section 4), coastal erosion (Section 5) and salt-water intrusion (Section 6).

## 2 Summary of previous assessments

An updated assessment of the impact and risks for natural and human systems by SLR is provided by the recent 6[th] Assessment
Report of the Intergovernmental Panel on Climate change (IPCC AR6), whose Working Group II (WG2, Pörtner et al., 2022) assessed impacts, adaptation and vulnerabilities related to climate change. SLR is considered in many chapters, particularly in chapter 3 "Ocean and coastal ecosystems and their services" (Cooley et al., 2022, see Cross Chapter Box 3 for an overall summary), and chapter 6 "Cities, settlements and key infrastructure" (Dodman et al., 2022), with more focussed material presented in the Cross Chapter Paper 2 "Cities and settlements by the sea" (Glavovic et al., 2022). Material directly addressing
European regional issues is included in chapter 13 "Europe" (Bednar-Friedl et al., 2022) and in the Cross Chapter Paper 4 "Mediterranean region" (Ali et al., 2022). The material on SLR covered by AR6 WG2 (Dodman et al., 2022) and the AR6 Synthesis Report (Calvin et al., 2023)) builds on and updates the former IPCC assessment in the "Special Report on the Ocean and Cryosphere in a Changing Climate" published in 2019, particularly in its Chapter 4 "Sea Level Rise and Implications for Low-Lying Islands, Coasts and Communities" (Oppenheimer et al., 2019).

There is extensive evidence that at global scale, relative SLR is already impacting ecosystems, human livelihoods, infrastructure, food security and climate mitigation potential at the coast (Pörtner et al., 2022). Observed impacts include chronic flooding at high tides, more frequent episodic flooding during storms, wetland, and underground water salinisation and ecosystem transitions, increased erosion and coastal flood damages. SLR poses risks for cities, settlements and populations in low-elevation coastal zones, cultural heritage along coasts, and threatens the very existence of some island nations. The
large exposure caused by the disproportional concentration of population and economic activities in coastal areas lead to high risks, which are very likely to increase further with future SLR. In addition, risks in the agriculture sector and nature conservation are induced by the salinisation of groundwater, estuaries, wetlands and soils. The IPCC AR6 WG2 shows that these risks will generally increase with SLR. At global scale and in the absence of further adaptation and mitigation action, they will be very likely one order of magnitude larger in 2100 than during present-day. Along many European coastlines,
extreme water levels, coastal floods and sandy coastline recession are projected to increase during the 21[st] century, because of the increase in relative SLR.

According to the IPCC AR6 WG2, there is high confidence that acceleration of SLR will increase risks to people and infrastructures from inundation and extreme floods along European low-lying coasts and estuaries. Related damages will
increase at least tenfold even before the end of the 21st century assuming present levels of adaptation and mitigation measures. Annual damage (which today is 1.3 billion EUR) is expected to increase disproportionately with global warming. It will be in the range 13–39 billion EUR by 2050 at global warming levels between 2°C and 2.5°C, and 93–960 billion EUR by 2100



between 2.5° and 4.4°C temperature increase. Assuming present distribution of population and protection levels, the increase of people at risk depends on the emission scenario: with respect to present, an extra 10 million people will be at risk of a 100-year flood event under a very high-emission scenario (RCP8.5) by 2100, whereas just below 10 million people under a low emission scenario (RCP2.6) by 2150. Along low-lying coasts and estuaries flood risks might further increase because of compounding storm surges, waves, rainfall, and river runoff events, but to date this has been poorly quantified (see also section 4). Port operations may be negatively affected by SLR in northern and western Europe. In Mediterranean ports negative effects are to be expected by a combined change of wave regimes and sea-level.

In the IPCC AR6 WG2 it is shown that soft cliffs and beaches in Europe are most affected by coastal erosion. Observations suggest that 27–40% of Europe's sandy coast are already eroding today. Though there is no convincing evidence that past erosion of sandy shorelines can be attributed to climate change or SLR. At the same time there is high confidence that SLR will increase sandy shoreline retreat in the future. Actual rates are uncertain, with large spatial variability and a strong dependence on local geometry, but it is suggested that by 2100 coastal retreat could reach approximately 100 m for a 4°C temperature increase, being reduced to less than 60 m under 3°C of global surface warming. On a centennial time-scale coastal erosion and flooding will become an existential threat for some coastal communities and UNESCO World Heritage sites. This is particularly the case in the Mediterranean region, where seawater intrusion in coastal aquifers is projected to increase by the combination of overexploitation and SLR, with pronounced impacts on agricultural productivity.

The IPCC AR6 WG2 shows that warming is the main climate hazard for European coastal ecosystems. However, rapid SLR (potentially aggravated by human induced subsidence) is also expected to have negative impacts by reducing the surface area of intertidal flats (e.g. the Wadden Sea) that cannot always be compensated by sediment accumulation. By 2100 under intermediate scenarios coastal erosion will cause the loss of 4.2–5.1% of present values of ecosystem services and reduce their contribution to shoreline protection across Europe. The vulnerability of Europe's coastal subtidal seagrass meadows and intertidal salt marshes to SLR is particularly high in the microtidal areas of the Baltic and Mediterranean coasts, with a potential loss of 75% of Posidonia oceanic seagrass habitats in the Mediterranean Sea.

## 3 The Source, Pathways, Receptors and Consequences framework

To describe the main impacts of SLR across Europe, we use the concept of Source-Pathway-Receptor-Consequence (SPRC), proposed by Sayers et al. (2002) as an alternative approach to the traditional, exposure vulnerability approach. The 'source' describes the origin of a hazard. The 'pathway' is the route that a hazard takes to reach the 'receptors', and includes processes and characteristics of the coastline influencing or mediating the hazard. The 'receptor' is the entity (e.g., people, property, environment) that may be harmed by the hazard, and the 'consequences' are the corresponding social, economic and environmental effects on the receptors. The concept is illustrated in Figure 2. The SPRC concept will be applied for coastal flooding (Section 4), erosion (Section 5), and salt-water intrusion (Section 6).



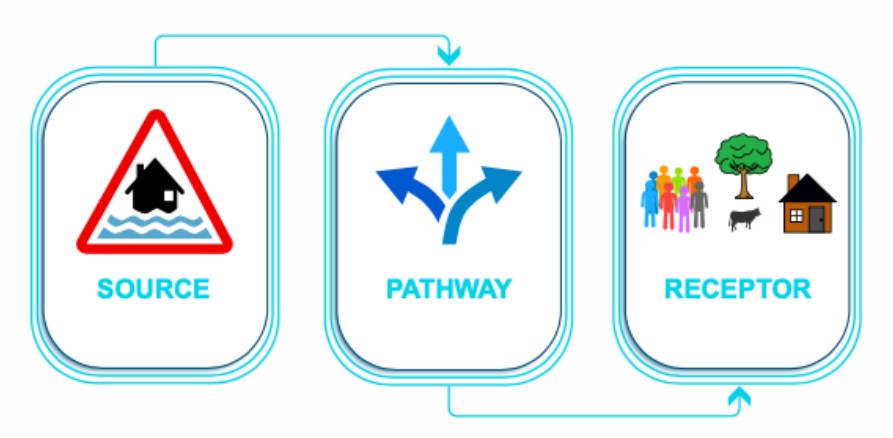


**Figure 2: Source-Pathway-Receptor-Consequence (SPRC) conceptual model (after Sayers et al., 2002)**

## 4 Coastal flooding and compounding flood events

The first pronounced impact of SLR we consider is coastal flooding. A schematic overview of the SPRC concept for coastal flooding is provided in Figure 3. Coastal floods are amongst the most impactful hazards both in Europe and globally, with
widely ranging social, economic and environmental consequences. Many severe flooding events have affected European coastlines throughout history (Ferrarin et al., 2022; Haigh et al., 2015, 2017; Melet et al., in prep.; Paprotny et al., 2018). An increase in coastal flooding frequency is one of the most certain and costly consequences of SLR (R. J. Nicholls & Cazenave, 2010). Flood-defense standards in many European countries are among the highest in the world (Galluccio et al., in prep.). However, significant populations and assets are in coastal flood plains and are threatened when defense infrastructure fails or
is exposed to flooding exceeding the protection standard. Furthermore, impacts of coastal flooding are likely to increase as flood defense infrastructure is ageing, as coastal population continues to grow, and urbanization and development in low-elevation coastal zones continue (Stevens et al., 2016; McMichael et al., 2020). In addition, further decline of the extent of natural habitats (i.e., saltmarshes, mudflats, shingle beaches and sand dunes), which act as a natural buffer to flooding, has the potential to increase coastal flood risks (Campbell & Keddy, 2022).




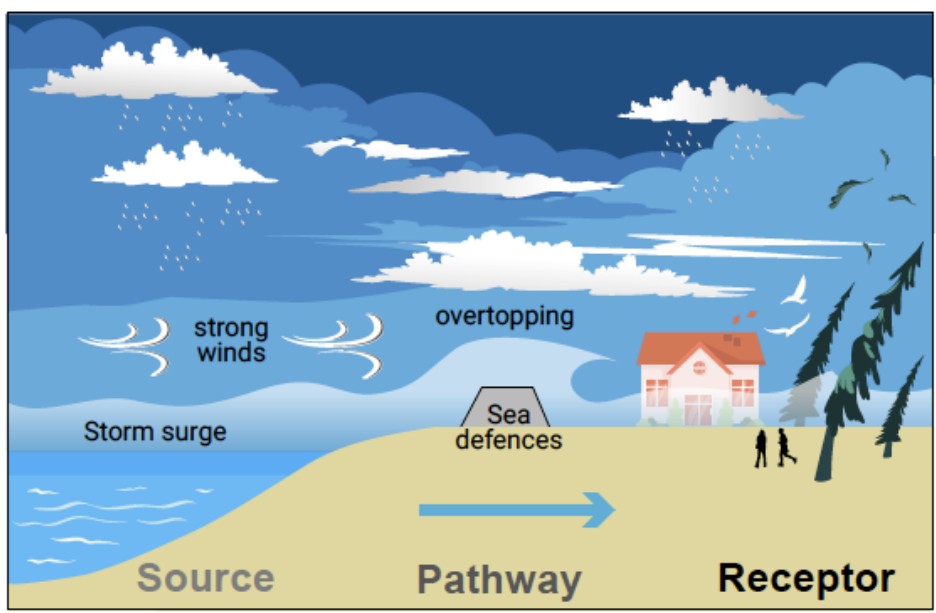

**Figure 3: The Source-Pathway-Receptor concept visualised for flooding.**

Multiple drivers, some related to climate change and others not, influence coastal flood risk and its future change. These different drivers can be considered using the SPRC conceptual model as explained in Section 3. The 'source' of coastal floods (describing the origin of governing hazards) is formed by extreme coastal water levels (including the contributions from tides, storm surge, wave runup, superimposed on relative mean sea level). The influence of climate change on the source component of coastal flooding is discussed in Section 4.1, and in sections 4.3 and 5.3 in (Melet et al., in prep.). In estuaries, the compounding influence of rainfall and fluvial input can be important leading to compound flooding, which is described as a special case in Section 4.2. The 'pathway' represents how seawater makes its way onto normally dry land to cause flooding. Climate change, especially SLR, can significantly affect this pathway in addition to modifying the source profile. This influence of SLR on pathways is discussed in Section 4.3. Receptors and consequences are closely linked and are discussed jointly in Section 4.4. Initiatives to develop flood-related climate services in Europe are discussed in Section 4.5.

**4.1 Source of flooding**

Coastal floods are governed by anomalously high-water levels exceeding a site-specific threshold. Extreme coastal water levels (ECWLs) arise as combinations of different drivers: (1) astronomical tides; (2) storm surges and associated seiches; (3) waves, including setup, infragravity waves and swash (e.g., (Dodet et al., 2019)); and (4) relative mean sea level (including SLR and land subsidence) (see also Sea Level Rise in Europe: observations and projections). These four components exhibit



considerable intra- and inter-annual variability (e.g., driven by tidal cycles (Haigh et al., 2011) or climate variability, such as the North Atlantic Oscillation (Boucharel et al., 2023; Hurrell, 1995; Mentaschi et al., 2017); or changes in wave climate (Morim et al., 2019; Melet et al., 2020). In addition, there are non-linear interactions between the four components (Horsburgh & Wilson, 2007; Idier et al., 2019; Arns et al., 2020). Long-term changes in any or all of the four components can modify the variability of frequency and magnitude of ECWL, and thus affect coastal flooding. The additional influence of rainfall and

fluvial input on water levels can be important in estuaries, discussed in Section 4.2 in relation to compound flood events.

ECWLs, and hence the frequency of coastal flooding events, are impacted by climate change in three main ways (Pugh & Woodworth, 2014) as also argued in Sea level rise in Europe: observations and projections: (1) SLR affects the ECWL directly, by raising the baseline mean water level leading to lower storm surge and/or wave elevations necessary to cause flooding; (2) rising mean sea levels alter water depths and therefore modify the propagation and dissipation of the tide and storm surge

components, or alter wave processes in shallow water (e.g., Arns et al., 2017; Chaigneau et al., 2023); and (3) climate variations in the tracks, speed, strengths and frequency of weather systems may alter the intensity and/or duration and frequency of storm surges and waves (and variations in rainfall and river discharge in estuaries).

As discussed in Sea level rise in Europe: observations and projections (Sections 4.3, 5.3), direct changes in mean sea levels appear to have been the main driver of observed changes in ECWL in the past (Ferrarin et al., 2022; Marcos et al., 2015;

Menendez & Woodworth, 2010) and are projected to dominate changes in extremes along the European coastline in the future, increasing the likelihood of coastal flooding (Vousdoukas et al., 2016, 2017). However, changes in storm surges (Calafat et al., 2022; Calafat & Marcos, 2020; Muis et al., 2020) and wave height (Aarnes et al., 2017; Benetazzo et al., 2022; Vousdoukas et al., 2018b) have and may also play a substantial role in the changes in ECWL in some European regions in the future. Coastal flooding could also be influenced by changes in tides, especially in regions with a shallow water depth. Regionally

coherent changes (positive and negative) in tidal range have been observed in historic sea level records around both European and global coastlines, and are projected to occur in the future with changes in water depth, driven by SLR and factors such as ice sheet extent and ocean warming (Ferrarin et al., 2015; Haigh et al., 2020; Idier et al., 2017; Pickering et al., 2012). Changes in tidal range are likely to be smaller than ±15% of mean SLR along most coastlines.

The relative contribution of SLR to changes in coastal flooding depends on different factors. Areas with tides below 2 meter

amplitude will be primarily affected by mean sea level changes. For example, in the Mediterranean Sea, extreme sea levels that are now occurring once in a century are expected to occur at higher frequency in future depending on climate scenarios but in some places even more frequently than annually. This intensification is stronger than the overwash and flooding in the South of Portugal, where current return periods of 1-in-100-year can reduce to lower than 1-in-20-year by 2055 and 1-in-10-year by 2100 (Ferreira et al., 2021). Venice stands as an exception for the Mediterranean Sea, as the existing protection

measures are higher than for most of the south European coastlines and could potentially protect the monumental city during the next few decades (Lionello et al., 2021; Mel et al., 2021).



### 4.2 Compound Flooding

Many coastal settlements along the European coastline are in estuaries and lagoons (e.g., London, Rotterdam, Hamburg,
Venice, Lisbon). In these coastal regions, floods can arise not only from climate driven changes in oceanographic sources (e.g.,
tides, storm surges and waves), but also via river discharge (fluvial) and direct surface runoff (pluvial). These mainly arise
from heavy precipitation, but are also incurred via snow melt. Many cities and towns located on the open coastline can also
experience flooding during heavy rainfall, because of insufficient drainage during high tides (Van Den Hurk et al., 2015a). In
the past, flood risk assessments typically considered the oceanographic, fluvial and pluvial drivers of flooding separately.
However, in coastal regions floods are often caused by more than just one factor, which can be physically correlated (e.g., with
storms). Furthermore, the adverse consequences of a flood can be greatly exacerbated when the oceanographic, fluvial, and/or
pluvial drivers occur concurrently or in close succession (i.e., a few hours to days apart), depending on local characteristics
which influence lag times between variables. This can result in disproportionately extreme events, referred to as 'compound
flood events'. (Zscheischler et al., 2018) define compound events as 'a combination of multiple drivers and/or hazards that
contributes to societal or environmental risk'. Flood drivers are typically causally related through associated weather patterns
(the modulator, Zscheischler et al., 2020) and therefore, it is assumed that stronger dependence between drivers increases the
impact of compound floods (Wahl et al., 2015).

In recent years there has been a large increase in the number of studies that have started to investigate compound flood events
in Europe. Many studies have been undertaken for specific localised regions in Europe, such as the Rhine delta, Netherlands
(Kew et al., 2013; Khanal et al., 2018); Brest, France (Mazas & Hamm, 2017); Santander, Spain (Rueda et al., 2016); Ravenna,
Italy (Bevacqua et al., 2017); Venice, Italy (Ferrarin et al., 2022); and the river Trent, the Yare basin, the river Ancholme, and
the rivers Taff and Lewes in East Sussex in the UK (Granger, 2001; Mantz & Wakeling, 1979; Thompson & Law, 1983;
Samuels & Burt, 2002; C. J. White, 2007). Larger-scale assessments of compound flood events have been undertaken more
recently for the UK (Hendry et al., 2019; Svensson & Jones, 2002, 2004) and for Europe (Bevacqua et al., 2019; Bevacqua et
al., 2020a; Camus et al., 2021, 2022; Paprotny et al., 2018; Petroliagkis et al., 2016). On a quasi-global scale, which obviously
includes Europe, Ward et al. (2018) and Couasnon et al. (2020) assessed the dependence between coastal and river flooding,
using observational datasets and reanalysis, respectively.

Most of these studies quantified the statistical dependence between flooding sources as an indirect measure of the flooding
hazard called compound flooding potential. The analysis of the interdependencies has been primarily limited to storm surge
and precipitation (compound surge-rain events, Wahl et al., 2015; Wu et al., 2018), or to surge and river discharge (compound
surge-discharge events, (Couasnon et al., 2020; Hendry et al., 2019; Ward et al., 2018). In those studies, precipitation is
considered as a fluvial proxy, which can be assumed as an equivalent driver to discharge in small to medium-sized river
catchments (Bevacqua et al., 2020b). Waves can be included as part of the total sea level, by adding it linearly to the storm
surge and/or astronomical tide components (Bevacqua et al., 2019).  Statistical studies have notably shown that: (a) the joint
exceedance probability of compound surge-discharge events  is on average a factor of 2-4 higher when the dependence is





considered (Santos et al., 2021; Van Den Hurk et al., 2015b; Ward et al., 2018); (b) ignoring the dependence between precipitation and surge can overestimate the flooding return period considerably (Bevacqua et al., 2019; Figure 4a); (c) the river discharge of the 50-year compound flood is up to 70% larger, conditional on the occurrence of extreme water levels (Ganguli & Merz, 2019a). Simulations of the non-linear interactions of these flood drivers at local scale using coupled

modelling approaches demonstrate a rise of extreme sea levels at some locations along the estuaries (e.g., in the Netherlands, Van Den Hurk et al., 2015; and in northwestern Spain, Bermúdez et al., 2021). Recently, a global analysis of simulated river flood levels using a global coupled river-coast flood model framework, showed that surge exacerbates 1-in-10 years flood levels at 64% of the river mouths analysed, with a mean increase of 11 cm (Eilander et al., 2020). Furthermore, 55% of the world coastlines face compound storm surge and wave extremes which increases the potential coastal flooding (Marcos et al.,

260   2019).

Although different sampling methods to identify compound events and dependence measures to quantify compound flooding potential have been used, several hotspots have been identified along the European coastlines. High joint occurrences of extreme river discharge and storm surge events are found on the coasts of Portugal, the Strait of Gibraltar, the western facing coasts of North and Central Europe (Heinrich et al., 2023), and along the south-west coast of the UK (Hendry et al., 2019).

Also, the northern and eastern Mediterranean coasts and the coast of Tunisia appear as compound flooding hotspots (Camus et al., 2021; Couasnon et al., 2020; Eilander et al., 2020). Regarding precipitation and surge, higher dependency is concentrated along the Atlantic coast and in the Mediterranean Sea (particularly in the regions of the Gulf of Valencia, Spain, northwest Algeria, the Gulf of Lion, France, the Adriatic coast of Balkan Peninsula, the Aegean coast, southern Turkey and the Mediterranean Levantine region) (Bevacqua et al., 2019; Camus et al., 2021, Figure 4a).

Historical trends in compound flooding resulting from high coastal water levels and peak river discharge have been assessed over north-western Europe (near the North and the Baltic Seas) over 1901–2014 using 37 stream gauges (Ganguli & Merz, 2019b). Increasing trends were identified in the region from 47°N to 60°N while decreasing trends were identified along higher latitude coasts (>60°N).

In most regional, continental, or global dependence-based analyses, compound events are identified using a two-sided

conditional sampling to bivariate drivers (Camus et al., 2021; Couasnon et al., 2020; Wahl et al., 2015; Ward et al., 2018) which implies that events are either conditioned to one driver or the other. Another approach is to select pairs of high values when both variables exceed individual high percentiles (e.g., 95[th] percentile, as in Bevacqua et al., 2019). However, extreme water levels might be driven by events not being extreme themselves. Impact-focused approaches, modelling the relationship between extreme water levels and underlying drivers, allow the selection of large impact events whose drivers are not necessary

extreme (Bermúdez et al., 2019; Bevacqua et al., 2017; Santos et al., 2021). Most of these studies rely on modelled data products which generally capture compound flooding, but contain biases, false positive or missed extremes (Paprotny et al., 2020).



Climate change can affect flooding dynamics through SLR, changes in each of the flood drivers and in the interaction processes between them. SLR can increase future compound flood hazard (Ganguli et al., 2020; Moftakhari et al., 2017). However, uncertainties due to internal climate variability and climate model differences dominate the large uncertainty in the concurrence of flood extremes in addition to changes of the drivers (e.g. storm surges or river discharge) themselves (Bevacqua et al., 2021). Large-scale sea-level/rainfall-driven compound flooding potential is projected to increase globally by more than 25% by 2100 under RCP8.5 scenario compared to present times (Bevacqua et al., 2020a). The probability of compound flooding due to the co-occurrence of high sea level and precipitation is projected to robustly increase by the end of the 21st century particularly in northern Europe, e.g., along the west coast of Great Britain, northern France, the east and south coast of the North Sea, but also along the coastlines of the eastern half of the Black Sea (Bevacqua et al., 2019, Figure 4b). Increases in compound flood events were observed to be mostly due to the rising mean sea level (Heinrich et al., 2023). These results contrast with a predominant decrease of compound flood hazard projected for north-western Europe at the middle of the 21st century (Ganguli et al., 2020), which may be caused by the fact that the two studies considered different flooding drivers and multi-model ensembles.

In terms of pathways for the impact of compound flooding, elevated sea levels can block or slow down river drainage into the sea, leading to increased upstream water levels which can overflow river channels onto adjacent floodplains thereby increasing inundation extent. Besides, floodwater can also erode riverbanks and cause landslides, leading to further flooding. Hydrodynamic modeling is required to provide a detailed spatial mapping of water levels in estuaries or coastal river deltas. However, high-resolution hydrodynamical modelling, which includes the non-linear interaction between hydraulic processes, topography, and human interventions is only feasible at local scale.

In terms of receptors and consequences, historical information on past damaging floods reveals that compound events have occurred in many locations in Europe. According to the HANZE database (Paprotny et al., 2018), out of 1564 floods that occurred in 37 European countries between 1870 and 2016, 23 (i.e. 1.5%) were identified to be compound floods, recorded in six different countries (the Adriatic Sea—Italian regions of Veneto and Friuli-Venezia Giulia, Mediterranean and western coast of France, Ireland, UK, Belgium and Poland). Specific examples of compound events include: flooding in December 2000 in Brittany, France – 600 persons were affected by a coastal flood that occurred due to the combination of heavy precipitation over several river catchments and a storm surge generated by an extra-tropical storm (Paprotny et al., 2018); and a flood in December 1999 in Lymington, UK – the town was flooded due to tidal locking of the high run-off down the Lymington River by a large surge produced by the same storm system, with this flooding overwhelming recently upgraded defences (Ruocco et al., 2011; Hendry et al., 2019).





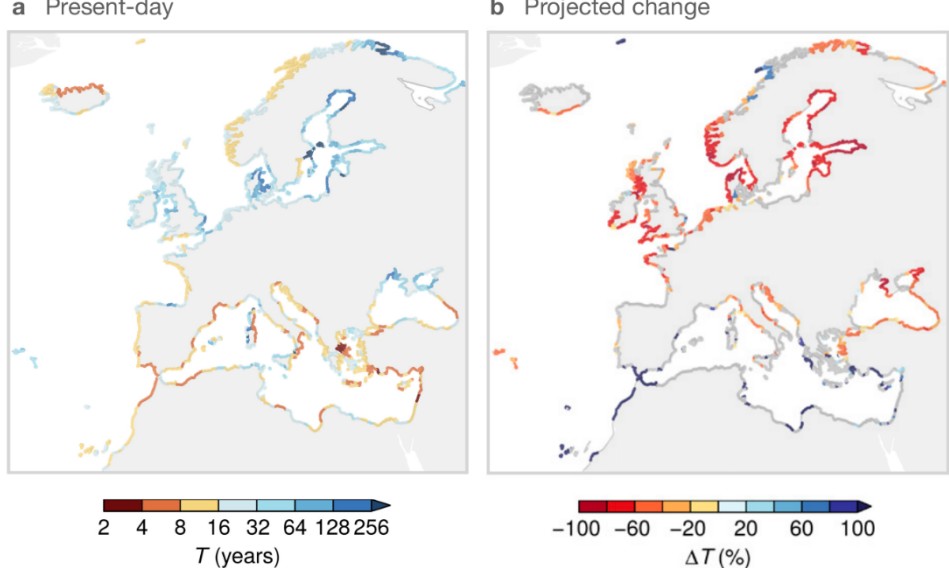

**Figure 4: (a) Present day (1980-2014) probability of potential compound flooding (CF). Return periods of CF (co-occurring sea level and precipitation extremes, i.e., larger than the individual 1-year return levels) based on ERA-Interim data. (b) Future probability of potential CF multi-model mean of projected change (%) of CF return periods, between future (2070-2099) and present (1970-2004) climate (Bevacqua et al., 2019).**

### 4.3 Pathway for coastal flooding

Seawater tends to inundate normally dry land via three main pathways: (1) by still water simply overflowing, where the water level exceeds the elevation of a natural (e.g., barrier beaches) or artificial (e.g., sea wall) barrier; (2) by waves overtopping a natural or artificial barrier; and (3) by breaching and lowering of a natural or artificial barrier, often as a consequence of prolonged overwashing or erosion at the front-face of the barrier or by groundwater seepage (Figure 3). Climate change, and other factors, can influence these pathways, altering local flood risk profiles. For example, SLR in regions with hard structures

like barriers or dykes typically lead to a decline in the extent of natural habitats, such as saltmarshes, mudflats, and sand dunes, which can act as a natural buffer to flooding (Hall et al., 2019). Decline in these natural features (and deterioration of flood protection infrastructure) can impact flood pathways and can increase flood hazard. In contrast, building new or maintaining and improving existing flood defences, or application of artificial nourishment and stabilisation of beaches, or providing more space for water through managed re-alignment, can alter flood pathways and reduce flood risk along coasts (Haigh et al.,

325 2022).

It is increasingly recognised that natural systems that provide important buffering against floods are in decline across parts of Europe. For example, Campbell and Keddy (2022) identified a 136 (confidence interval 39–236) km$^2$ loss of salt marsh extent from 2000 to 2019 across Europe. Other green adaptation options include restoration of seagrass meadows (which reduce wave height and sediment erosion) and the creation of buffer zones (Wolff et al., 2023). With mean SLR accelerating over the





coming century, and thereby increasing pressure on the narrow coastal zone there is likely to be a continued decline in the extent of natural systems contributing to natural buffers reducing coastal flood risk, which will lead to defence capital and maintenance costs increasing dramatically (Haigh et al., 2022). This is corroborated by projections of shoreline retreat along most of the global shorelines induced by SLR (Vousdoukas et al., 2020; Section 5) and a consequent reduction of ecosystem services (Paprotny et al., 2021).

Obviously, current flood risk around the coastline of Europe would be considerably higher without the decades of investment into extensive flood risk management infrastructure. Data on flood defences over time are not well documented, but massive investments in defences have occurred over the 20th and early 21st century in Europe. For example, extensive flood defence infrastructure has been built in the Netherlands as part of the Delta Works and standards of protection along stretches of the Dutch coastline now reach 1-in-10,000-year levels (Eijgenraam et al., 2014). Governing policy directives incorporate future

sea level rise into the periodic risk assessments and defence strategy updates (Kothuis and Kok, 2017). Nearly a quarter of England's coast is now defended (Sayers et al., 2015). As for France, more than 16,000 coastal works and developments are covering more than 2,300 km of coastlines, implying 17% of metropolitan French coasts are shaped by human interventions. Around 20 movable storm surge barriers have been built around the coast of Europe since 1958 offering flood protection to millions of people and trillions of Euros of infrastructure (Mooyaart and Jonkman, 2017). This includes six surge barriers in

Netherlands (e.g., Eastern Scheldt and Maeslant Barrier, Figure 5), thirteen in the UK (e.g., Thames and Hull Barriers), and one in Italy (the MOSE system for protecting the historical city of Venice and the lagoon settlements, operating since October 2020), while a new surge barrier is being constructed in Belgium (Nieuwpoort).  However, storm surge barriers can be used to mitigate the flood impact only in semi-enclosed coastal environments (such as bays, estuaries, lagoons) but not along coasts facing the open sea. Extensive beach nourishment has also taken place in many European countries to counteract coastal

erosion and flooding. For example, the Dutch coast is one of the most heavily nourished coasts globally (Brand et al., 2022); since 1990, more than 300 nourishments programs have taken place (including the notable Sand Engine project; Roest et al., 2021) adding an average of 12 million m$^3$ annually to the 432 km of Dutch coastline.

However, as sea-levels continue to rise it will become increasingly costly to maintain existing flood defences and surge barriers, and to carry out coastal nourishments. In the UK, Sayers et al. (2015) showed that length of coastal defences 'highly

vulnerable' to failure would almost double (triple) under 0.5 m (2.5 m) mean SLR, with the number of properties affected rising by around 160% (490%). Furthermore, many of the existing storm surge barriers will need to be strongly upgraded or replaced over the coming century as sea-level rises. For example, plans are underway to replace the Thames Barrier around 2070, with options including a new barrier built farther downstream of the current one (Environment Agency, 2021). The Eastern Scheldt Barrier (Figure 5) was designed for only 40 cm SLR, implying that a revision is needed to avoid it to be closed

too often with higher sea-level (Haasnoot et al., 2020) with as consequence that the availability of maintenance windows decreases too much.



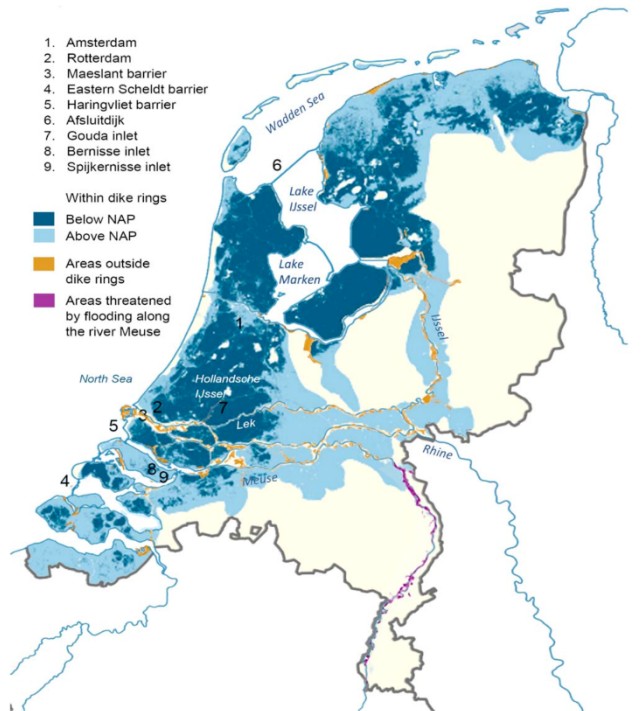

Figure 5: Map of the Netherlands showing flood prone zones (blue shadings) and features of the water management system. NAP is the Amsterdam Ordnance Level which is the reference plane for sea level height in the Netherlands. Extracted from Haasnoot et al., 2020.

## 4.4 Receptor for and Consequences of flooding

The receptors and consequences associated with coastal flood events can be broadly grouped into social (e.g., loss of life, number of people evacuated, damage to residential property, or loss of cultural heritage), economic (e.g., overall monetary cost, disruptions to ports, transport, energy, public services, water systems, agricultural production losses) and environmental (e.g., coastal erosion, degradation or losses of coastal habitats, damage) impacts (Haigh et al., 2017). Importantly these consequences can be long-lasting (e.g., injury or long-term physical and mental health effects; Jackson and Devadason, 2019) and can also extend outside of the coastline area directly impacted by flooding (e.g., disruption to transport or supply chains; Dawson et al., 2016). As SLR and increases in storminess enhance flood risk and its consequences, the number of receptors in flood-prone areas will grow accordingly. Changes in land use and increasing asset values in floodplains can also increase consequences of coastal flooding (Haigh et al., 2022). In contrast, improvements in flood forecasting, early warning, emergency response and planning can greatly reduce the consequences of flooding (Section 4.5). Careful spatial planning and building codes can be effective at reducing risk. Evidence from Haigh et al. (2017) suggests that the number and consequences of coastal floods have declined since 1915 in the UK, reflecting better defences and improvements in flood forecasting,



warning, emergency response and planning (Haigh et al., 2022). As a concrete example, more than 2000 lives were lost around the coastlines of the North Sea during the flood of 1953; however, similar conditions occurring in December 2013 hardly had any societal impact, which can be attributed to these improvements and infrastructure investments (Wadey et al., 2015).

As shown in Figure 6, the flooded area from the 100-year event is projected to increase by 10%-15% in 2050, depending on the emissions scenario, and by 12%-20% by the end of the century. For more rare events, like a 1000-year event, SLR will

drive a 48%-67% increase in the flood extent by the year 2100 (Paprotny et al., 2019). The country level flood extents depend on different factors, such as the exposure to extreme weather conditions, the protection standards in place, as well as the country's size and the percentage of low-lying coastal areas. For the baseline, the UK and Norway have the largest flood extent area, exceeding 4,500 km$^2$ for the 100-year event for each country. This is a result of their long coastlines exposed to intense weather conditions. Denmark and Germany follow with values around 3,000 km$^2$, and for these cases the main driver is the

flat and low-lying configuration of the coastal zones. Other countries with flood extents slightly below 2,000 km$^2$ are Greece and Italy, both characterized by long coastlines.





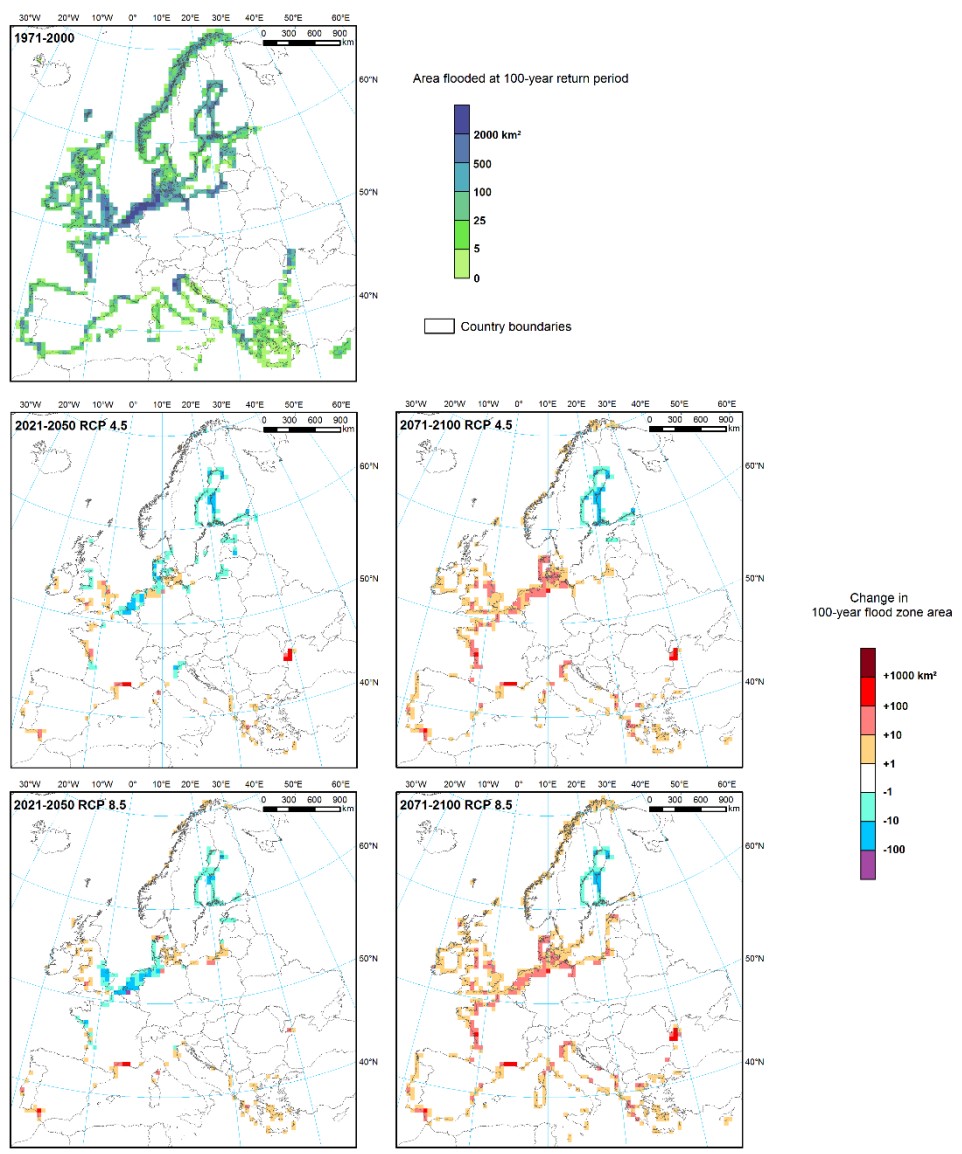

**Figure 6: Indication of the current area vulnerable for flooding with a return period of once in a hundred years and for two different time slices under two different climate scenarios. From Paprotny et al. (2018).**

It is noteworthy that also events with low flood levels (i.e., nuisance flooding, Moftakhari et al., 2018) may have high impacts from disruption of everyday routine activities and property damages, especially in low land settlements. SLR is expected to increase not only the frequency of ECWL but also the frequency of low-level floods, when astronomic tide will become a major driver of flood events (Ferrarin et al., 2022). Towards the end of the century SLR will also result in permanent flooding of certain areas. For example, in the Balearic Islands and by the year 2100, 7.8–27.7 $km^2$ and up to 10.9–36.5 $km^2$ will be permanently lost to SLR under RCP4.5 and RCP8.5 emission scenarios, respectively (Luque et al., 2021).



Pan-European assessments of future coastal flood risk show general trends and allow for regional comparisons, but also come with large uncertainty due to data scarcity and non-stationary conditions (Hinkel et al., 2021; Vousdoukas et al., 2018a) and

cannot replace local studies (Paprotny et al., 2019). For example, flood assessment results are very sensitive to the coastal protection standards assumed which are largely unknown along most of the European coastline (Scussolini et al., 2016). Another crucial factor is digital elevation data. Since several countries lack high-resolution LIDAR data, many assessments are based on less accurate global datasets, which come with vertical biases exceeding the extent of anticipated SLR (Bove et al., 2020; Kulp & Strauss, 2018; Yamazaki et al., 2017). Despite these limitations, all known flood risk assessments highlight

an increase in flooded area during the century, which also accelerates as SLR gathers pace.

### 4.5 Initiatives to develop flood-related climate services in Europe

The EU Adaptation Strategy emphasises the significance of climate services in adapting to climate change. As defined by the Global Framework for Climate Services (Hewitt et al., 2012) of the World Meteorological Organization (WMO), "Climate

services provide climate information to help individuals and organizations make climate smart decisions". According to the European Commission's Roadmap for Climate Services (European Commission, 2015) definition, climate services cover "the transformation of climate-related data - together with other relevant information - into customized products such as projections, forecasts, information, trends, economic analysis, assessments (including technology assessment), counselling on best practices development and evaluation of solutions and any other services in relation to climate that may be used for the society

at large".

Information on past and future sea level change close to the coastline is being made available, but still suffers from various limitations (e.g., resolution, incomplete physics in models, scarcity of observations). In parallel, core European services have been put in place. Yet, authoritative, consistent and decision- oriented climate services to support policies and decision-making with SLR are still in their early development worldwide.

The European Union's Earth Observation Programme (Copernicus) monitors our planet and its environment, for the ultimate benefit of society. This includes the monitoring of sea level changes and the provision of ancillary fields needed to assess coastal SLR risks, as well as the transformation of the wealth of satellite, in situ and integrated numerical model information into added-value datasets and information usable by scientists, managers, decision-makers and the wider public to guide adaptation and to support related policies and directives (Melet et al., 2021).

New initiatives in the framework of dedicated European research projects in the Horizon Europe program are also underway, for instance the Coastal Climate Core Services (CoCliCo) project. CoCliCo aims to deliver an open web platform that will provide up-to-date information on present and future SLR and its impacts to support decision-making on coastal flood risk management and adaptation. The platform will grant access to the latest and consistent hindcast and projections of sea level, process-based coastal flood maps and shoreline change estimates, flood exposure and vulnerability information, as well as

adaptation strategies and options. Users of the platform will be able to visualize, download, and analyse high-quality geospatial



information layers encompassing multiple decision-oriented coastal risk scenarios. For shorter timescales, Early Warning Systems (EWSs) are integrated systems allowing a real time monitoring of potential natural hazards, issuing natural hazard warnings with a few days lead time. Informing the relevant stakeholders (e.g., civil protection agencies, regional and local authorities, environmental agencies) is part of an integrated risk management procedure to mitigate risks. EWSs can play a

critical role in classical disaster risk management cycles, supporting the preparedness and response phases, including the deployment of emergency measures for rapid response after a disaster, as well as longer term damage assessment after the occurrence of an event. EWSs are an efficient adaptation measure by providing more than a tenfold return on investment (Global Commission on Adaptation, 2019). The H2020 ECFAS project aims at contributing to the evolution of the Copernicus Emergency Management Service (CEMS) by demonstrating the technical and operational feasibility of a European Coastal

Flood Awareness System. Such a system will complement the existing early-warning system of the CEMS for river/pluvial floods, CEMS-EFAS (European Flood Awareness System), by adding a pan-European marine coastal flood awareness system and by tackling coastal resilience to climate risk (marine storminess and exposure). ECFAS provides an integrated risk cycle monitoring and management service, from water level forecasts at the coast with a 5-day lead time (Irazoqui Apecechea et al., 2023), rapid mapping of coastal floods and impacts on population and assets (Le Gal et al., 2022), and risk and recovery

Mapping (RRM) for adding coast-targeted products in the aftermath of a marine flood event (e.g., shoreline displacement, maps of flooding and damages). At national scale, the Norwegian mapping authority developed a webtool for inundation mapping showing extreme still water levels and projected sea level, including statistics on the areas, roads, and buildings affected now and in the future (Breili et al., 2020, https://www.kartverket.no/en/at-sea/se-havniva/se-havniva-i-kart). The webtool includes statistics on the areas, roads, and buildings affected now and in the future. The German Sea Level Monitor

(https://meeresspiegel-monitor.de/index.php.en) provides observed and projected changes at tide gauges along the German North Sea and Baltic Sea coast.

## 5 Coastal erosion

### 5.1 Definition and drivers of coastal erosion

Coastal erosion is the permanent loss of land to the sea. Coastal erosion can take place at different time scales and is driven by

a wide range of natural and anthropogenic factors. Among the different types of coastal systems are estuaries, lagoons, barrier islands, sandy and gravel beaches, dunes, cliffs, rocky shores, and built areas. Sandy beaches are especially prone to erosion (EUROSION, 2004a, 2004b). They occupy 31% of the ice-free global coastline (Luijendijk et al., 2018) and tend to be the most heavily utilized beach typology (Davenport & Davenport, 2006). Sandy beaches also tend to be geomorphologically complex and dynamic, affected by natural and anthropogenic factors (Mentaschi et al., 2018). A large part of the European

coastline is developed by humans, and the built area deprives coastal systems from their natural capacity to accommodate, or recover from erosion (Small & Nicholls, 2003). In addition, human developments and dams tend to prevent terrestrial sediments from reaching the coastline, which favours coastal erosion (Milliman, 1997). As a result of the above, Europe's





beaches have been eroding (EUROSION, 2004a, 2004b; Masselink et al., 2022; Romagnoli et al., 2022), a trend which is projected to accelerate with climate change and SLR (Vousdoukas et al., 2020).

One of the consequences of erosion is shoreline change which is the combined result of numerous factors, such as wind and wave climates, terrestrial sediment supply, geological control and human interventions, among others. There is a clear cause and effect relationship between increasing sea levels and shoreline retreat (Bruun, 1962), which justifies the eroding trend reported by either large scale (Hinkel et al., 2013; Vousdoukas et al., 2020), regional (Alvarez-Cuesta et al., 2021; Toimil, Losada, Camus, et al., 2017), or local scale studies (Alvarez-Cuesta et al., 2021; de Santiago et al., 2021; Luque et al., 2021;

Romagnoli et al., 2022; Toimil, Losada, Díaz-Simal, et al., 2017). Negative sediment budgets can be another factor driving a robust erosion trend (López-Olmedilla et al., 2022). The former can be the result of several potential natural or anthropogenic factors. In addition, several of the observed changes can relate to quasi-periodical climatic patterns affecting the wave regime (e.g., Barnard et al., 2015).

Apart from major drivers like the ones mentioned above, beach erosion and accretion often depend on a delicate balance of

contrasting forces with very similar amplitudes and are therefore very difficult to predict. This applies to all scales; i.e. each wave can transport sand both towards and away from the coast (Vousdoukas et al., 2014), while storms drive erosion that can be followed by complete or partial recovery (e.g., Kroon et al., 2008; Lee et al., 1999). Moreover, the impact of extreme storm events is very much controlled by the initial beach morphological state, and thus by meso-to-macro scale processes (e.g., Qi et al., 2010; Vousdoukas et al., 2012). In this Section, the source, pathway, receptor and consequence are discussed for coastal

erosion in Section 5.2. Monitoring methods, including field surveys, video monitoring and Earth observation, are discussed in Section 5.3. Historical shoreline changes in Europe are summarized in Section 5.4 while projected shoreline changes are summarized in Section 5.5. Finally, interactions between coastal erosion and coastal flooding are addressed in Section 5.6.

**5.2 Source/Pathway/Receptor/Consequence for coastal erosion**

*Source for erosion*

Coastal erosion shares the same sources as coastal flooding; e.g., all the components that drive ECWL (tides, storm surges, waves and SLR) can drive coastal erosion, Figure 7. In addition, given that coastal erosion is strongly related to the sediment budget, sediment sources and sinks are very important. As a result, human development and dams play an important role as they tend to prevent terrestrial sediments from reaching the coastline favoring coastal erosion (Anthony et al., 2019; Milliman,

495 1997).



**Coastal Erosion**

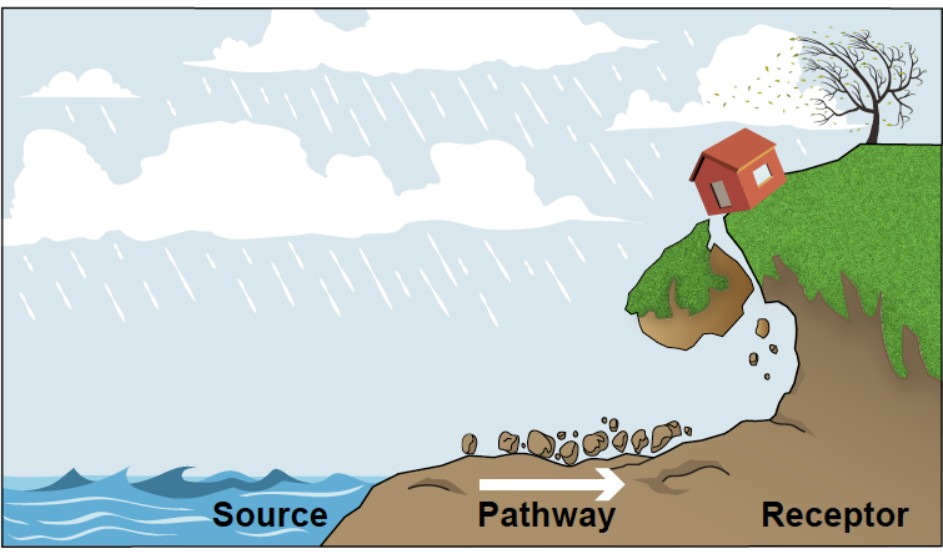

**Figure 7 Source, Pathway, Receptor Consequence framework for coastal erosion.**

*Pathway for erosion*

Coastal erosion takes place when the sediment budget of a given area becomes negative. Understanding coastal change is a very challenging task since erosion or accretion is a result of multiple factors like: (i) processes that take place in various temporal and spatial scales (Kroon et al., 2008; Larson & Kraus, 1994); (ii) the presence of various features which emerge through complex, self-organization processes often linking different scales and processes (Murray et al., 2009; B. T. Werner, 1999); (iii) the intrinsic uncertainty in predicting the intensity and frequency of extreme events, as well as the related beach
morphological response (Vousdoukas, 2012); and (iv) the high complexity of long-term processes like sediment transport and vertical land motion, which are also interconnected, among others, with geological and meteorological phenomena (Gallop et al., 2011; Vousdoukas et al., 2007), as well as human interventions which are increasing in number and extent (Luijendijk et al., 2018; Mentaschi et al., 2018). In principle, coastal erosion can be the result of any process that alters the sediment transport patterns. This can be either hydrodynamic (e.g., changes in wave intensity or direction, sea level, etc) (Sierra & Casas-Prat,
2014), related to the presence of obstacles like hard structures (Loureiro et al., 2012; Noble, 1978), or factors that affect the erodibility of the beach (Feagin et al., 2019).

*Receptor and consequences for erosion*

   Coastal erosion is the process by which the land is worn away and is submerged in water. Coastal erosion can take many forms,
such as the loss of sand dunes, cliffs, or beaches and can have several consequences. Among them are the destruction of buildings, roads, and other infrastructure located near the coast.



Moreover, coastal erosion can have significant impacts on the environment, economy, and human health and safety, as it can lead to habitat loss for coastal species, reduce the recreational value of beaches, and increase the risk of flooding and storm damage for coastal communities.


### 5.3 Monitoring methods for coastal erosion

Coastal monitoring is crucial to gaining a better understanding of the likely impacts of climate change at the coast. While the long-term implications of SLR have received considerable attention, Nicholls et al. (2007) point out that more attention needs to be given to finer temporal and spatial scales, including the localized impacts of potential changes in wave climate and

storminess regimes. However, monitoring the coast at higher temporal (daily to decadal) and spatial (three-dimensional) resolutions presents many challenges. Conventional survey techniques for ongoing beach surveys are both costly and labor-intensive, and meaningful trends typically require several years of data to emerge (Short & Trembanis, 2004). Currently, only half a dozen multi-decadal, high-resolution coastal monitoring programs are in operation worldwide, with only 2 in Europe: Truc Vert (France) since 2005, and Noordwijk (Netherlands) since 1964.

The various monitoring methods for coastal erosion are described briefly in Table 1, highlighting the evolution from traditional to present-day monitoring techniques.

**Table 1: Summary of the methods for monitoring coastal erosion**

| Type of method | Brief description of method |
| --- | --- |
| Field surveys | GPS surveys have proven to be a successful method for beach topographical profiling on sandy coasts (Hansen & Barnard, 2010; Harley et al., 2011). This technique is commonly used for monitoring morphological changes in the short term and involves repeating measurements at regular intervals to understand the physical aspects of coastal environments, including daily, monthly, and annual variations of specific parameters (Komar, 1998; Short, 1999). Spatial scales of such surveys are typically from 100m's to several km's. |
| Video monitoring | Coastal video monitoring systems (e.g. Holman and Stanley, 2007; Stringari and Power, 2022) are ideal for long-term deployments (i.e., years) acquiring data continuously, covering few km of coastline. Images are processed to generate the system's 'basic products': time-averaged, variance, snapshot and timestack images, which are all projected in geographic coordinates using standard photogrammetric techniques. Then, a set of post-processing tools allows extracting quantitative information on various coastal processes at various temporal and spatial scales, such as beach-face/shoreline morphology, inner bar configuration, rip/longshore current systems and wave run-up. Moreover, they have been proved to be useful coastal management tools, supporting sustainable and safe recreational beach use (Jiménez et al., 2007). |





| Drones | For the past 10 years, drones or Small Unmanned Aerial Vehicles (e.g. Vousdoukas et al., 2011) have been used more frequently for coastal monitoring (Chapapría et al., 2022). These unmanned aerial vehicles are equipped with cameras and other sensors that allow researchers to collect high-resolution images and data over km's of the coastline. This technology offers several advantages over traditional methods of monitoring. Drones can cover a larger area than ground-based surveys and provide more detailed images than satellite imagery. Additionally, drones can be deployed quickly and easily, making them ideal for collecting data in remote or hard-to-reach areas. |
|---|---|
| Terrestrial 3D Laser Scanning | Terrestrial 3D Laser Scanning (TLS) or Terrestrial LIDAR has increasingly become the method of choice for beach surveying (e.g., Pietro et al., 2008). Portable scanners allow the completion of beach surveys in excess of hundreds of meters, with sub-centimeter spatial resolution, during the course of a few hours. These advances provide a reliable means of addressing geomorphic relationships, as well as along- and cross-shore changes on the beach; while the accuracy and spatial resolution is practically impossible with traditional GPS surveys. The technique has been very recently introduced in coastal research and has several unexploited possibilities. |
| Earth Observation | In recent years a new source of geospatial data for studies from regional to planetary scale is provided by earth observation satellites generating an ever-increasing flow of raster image data. An exponential increase in the availability of free geospatial data has recently emerged with the Copernicus Earth observation and monitoring program of the European Union that delivers satellite imagery complemented by in-situ observations (Malenovský et al., 2012).<br><br>The positional accuracy of satellite derived shorelines (SDS) based on single images has been evaluated to range between 1.6 and 10 m (e.g. Liu et al., 2017). The increasing availability, resolution and spatial coverage of satellite imagery in recent years now provide a powerful alternative to derive reliable, global scale shoreline data. |


## 5.4 Historical shoreline change

A comprehensive European study of coastal erosion collected and analysed aerial photographs and local surveys up to 2002 to estimate coastal erosion at the scale of Europe (EUROSION, 2004a). About 20% of the European Union's coastline suffered serious erosion impacts, with the area lost or seriously impacted estimated at 15 km$^2$ per year. More recent studies at continental

to global scale confirmed that large stretches of the European coast are suffering from erosion. Using freely available optical satellite images captured since 1984, in conjunction with machine learning and image analysis methods, the shoreline changes have been mapped at global scale (Luijendijk et al., 2018; Mentaschi et al., 2018). The application of an automated shoreline detection method to the sandy shorelines resulted in a global dataset of shoreline change rates for the 33-year period 1984–2016 (Luijendijk et al., 2018). Analysis on satellite-derived sandy beach detection reveals that about 35% of the European





coastline is sandy, which agrees largely with the 40% Eurosion estimate (EUROSION, 2004a). Analysis of the satellite derived

shoreline data indicates that 22% of the European sandy beaches are eroding at rates exceeding 0.5 m/yr, while 26% are

accreting and 52% are stable. This means that in Europe a total of more than 8,200 km of sandy beaches have significantly

retreated over the last decades. Areas that experience severe erosion are found at various locations across Europe (see Figure

8). About 4% of the European sandy beaches experience erosion rates classified as severe (> 3m/year). Erosion rates exceed

5 m/yr along 2% of the sandy shoreline. The statistics of Europe are rather similar to the global statistics stating that 24%

(28%) of the world's beaches are eroding (accreting) (Luijendijk et al., 2018).

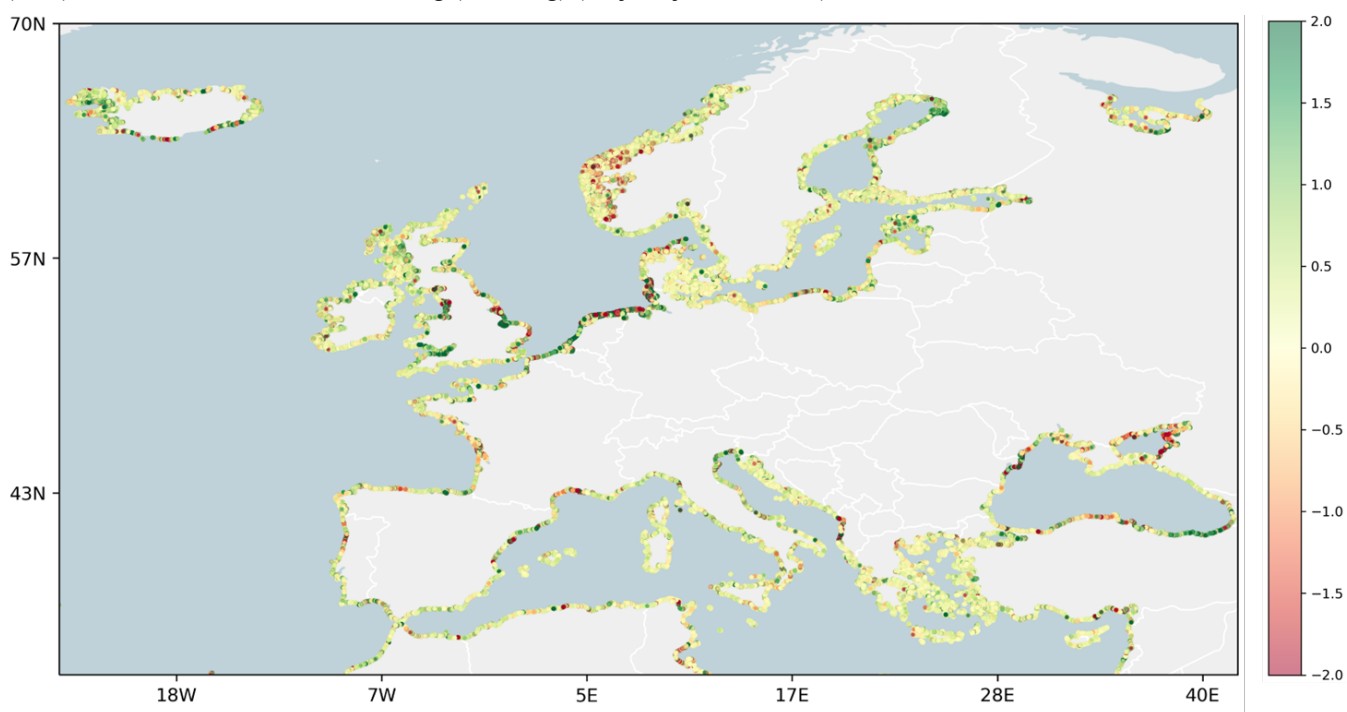

**Figure 8: Long-term shoreline changes of the European beaches in m/yr since 1984. The red and green points indicate erosion and accretion respectively (data based on Luijendijk et al. (2018)).**


**5.5 Future shoreline change**

Mediterranean beaches are more susceptible to the negative effects of SLR because they are narrower as a consequence of the

beach slope. This is highlighted by both large scale (Vousdoukas et al., 2020) and regional scale projections (Monioudi et al.,

2017). For example, a recent study in the Balearic Islands projects at least 20% of the islands' beaches losing more than 50%

of their surface by the end of the century, even if greenhouse gas emissions are mitigated (Luque et al., 2021). But even

projections along the Atlantic coast report shoreline retreat; e.g., in the range of 10-45 m under the middle-of-the-road

(RCP4.5) scenario and 14-66 m under the very-high emission (RCP8.5) scenario by the year 2100 for 150 km of Basque coast

(de Santiago et al., 2021). A recent pan-European study projects a mean SLR driven median shoreline retreat of 97 m (54 m)




under RCP8.5 (respectively under RCP4.5) by the year 2100. This retreat translates to 2,500 km² (1,400 km²) of SLR driven

coastal land loss (Athanasiou et al., 2020), Figure 9. Given the complexity of coastal morphodynamics, projections of shoreline

changes come with high uncertainties. The future variability of wave forcing is more prominent until 2060 with respect to

uncertainties, whereas after that year the uncertainties in predicting sea-level rise become dominant (D'Anna et al., 2021,

2022). Of similar amplitude is also the uncertainty associated with the choice of geophysical datasets in continental scale

assessments (Athanasiou et al., 2020), reaching 45% (26%) of the variance in coastal land loss projections for Europe by 2050

570    (2100).

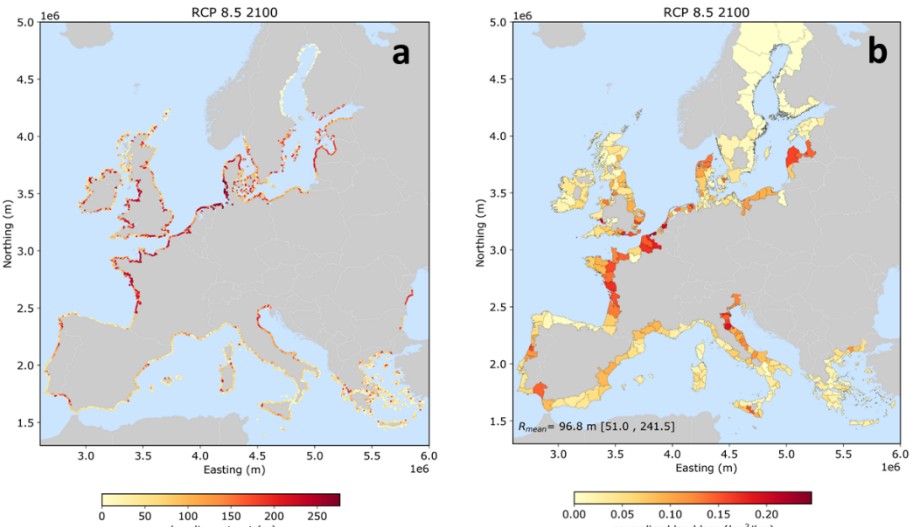

**Figure 9: Projections of shoreline retreat in m (a) and land loss in km² per km of coastline (b) for Europe in the year 2100 under a very high emission scenario (RCP8.5) (from Athanasiou et al., 2020).**


**5.6 Coastal erosion and flooding interactions**

Coastal erosion and flooding are extremely interrelated impacts that influence each other (Leaman et al., 2021; Pollard et al.,

2019; Sallenger, 2000). Erosion is a physical phenomenon where sand is removed from the shoreface and deposited elsewhere,

usually offshore. Erosion and deposition processes can change the shoreface, which affects coastal flooding caused by high

water levels. In turn, these high-water levels can cause further erosion or deposition. This feedback manifests itself on different

time scales. In the short term, coastal morphology plays a significant role in wave energy dissipation, total water levels reaching

the coast, dune breaching, and subsequent flooding. SLR is expected to increase the frequency of episodic erosion and flooding,

which in turn could be altered by changes in storminess. At longer time scales, SLR is expected to drive permanent erosion

and inundation (R. J. Nicholls & Cazenave, 2010; Cazenave & Le Cozannet, 2014). This effect could be compensated or

enhanced in areas with intense alongshore gradients in longshore sediment transport or chronic fluvial sediment supply. Higher





water levels cause wave-driven erosion to occur higher up in the profile resulting in net erosion and deposition on the nearshore bottom (Bruun, 1962). Additionally, deeper water reduces wave refraction and allows waves to get closer to the shore before breaking (Arns et al., 2017; Chaigneau et al., 2023), leading to increased flooding. A coastline subject to sustained erosion over time may lead to the loss of natural flood defenses (Toimil, Losada, et al., 2023; Toimil, Álvarez-Cuesta, et al., 2023).

The need for the joint modelling of flooding and erosion has long been recognized in the literature (Bilskie et al., 2014; Lentz et al., 2016; Passeri, Hagen, Bilskie, et al., 2015). However, most studies to date continue to analyse these two impacts separately because of the complex relationship between driving processes and morphological response (Pollard et al., 2019; Toimil et al., 2020; Toimil, Losada, et al., 2023). Amongst the studies that couple flooding and erosion, most of them have typically focused on historical events without considering climate change (Gharagozlou et al., 2020; McCall et al., 2010; Van

Ormondt et al., 2020). These studies have primarily been conducted at the storm scale and have used pre- and post-storm topo-bathymetry data to simulate erosion and breaching using a hydro-morphodynamic model. Following the same modelling approach but in the context of climate change, some studies have incorporated the effect of SLR on hydrodynamic forcing conditions (Grases et al., 2020; Passeri et al., 2018). Sanuy and Jimenez (2021) presented a more recent application in the Tordera Delta (Spain) where the baseline topo-bathymetry was modified to consider medium-term erosion.

Studies that consider long-term shoreline changes in flooding follow more diverse approaches. Stripling et al. (2017) delivered flood maps considering long-term changes in seawall toe levels along the west coast of Calabria (Italy) and in an idealised coastal stretch around Holderness (UK). As for climate change studies, Dawson et al. (2009) developed a methodology to account for shoreline changes in coastal flood projections along the East Anglian coast (UK). Dawson et al. (2009) linked coastal flooding due to storms and SLR with long-term erosion by adjusting the likelihood of flood defence structures failure

based on shoreline changes. Other studies have examined the effect of shoreline changes on coastal flooding considering empiric (Grilli et al., 2017) and surveyed profile translation due to SLR (Barnard et al., 2019) and due to SLR and changes in the sediment budget (Passeri et al., 2016; Passeri, Hagen, Medeiros, et al., 2015). Also using real profiles (Toimil, Álvarez-Cuesta, et al., 2023) proposed a suite of numerical and statistical models to analyse the influence of storm morphodynamics, SLR, erosion and longshore sediment transport on total water levels and coastal flooding along a 40-km coastal stretch in the

Spanish Mediterranean.

Current studies highlight the need to consider the interconnections between hydrodynamics and morphodynamics to better understand the functioning of the coastal system. Changes in topographic representations can alter the path and pattern of maximum water levels (Bilskie et al., 2014). (Toimil, Álvarez-Cuesta, et al., 2023) found that total water levels are mainly affected by storm erosion and profile geometry, and that long-term erosion is the main shoreline change contributor to the

flooded area.

To date, existing research in the EU focuses on the East Mediterranean and North Sea basins. In the East Mediterranean, studies encompass a wider range of coastal typologies than in the North Sea, including cliffs, deltas, and beaches with varying levels of anthropisation. Most studies model the interaction of medium and long-term coastal changes with episodic flooding,



but few additionally consider storm erosion as flood enhancer. This can be particularly important for extreme weather events
of high return periods, regardless of the type of coastline. The largest gap lies in studies that combine the interaction of hydro-
morphological processes under climate change scenarios, as such studies are very limited in both basins. However, knowledge
on the potential effects of these interconnections and the enhancing role of SLR is key information to decision makers to make
informed decisions for the future management of coastal communities.

## 6 Saltwater intrusion

### 6.1 Processes and monitoring of saltwater intrusion

Saltwater intrusion (SWI) represents the increased extent of the mixing zone between inland freshwater and saltwater (Source),
therefore rising the salt content both in surface waters and in groundwaters (Pathway). SWI can hinder the use of water for
agriculture due to salt damage to crops (e.g., Maas and Hoffman, 1977) and can threaten coastal communities that rely on fresh
water supplies for their livelihood (Receptor) (e.g., too salty drinking water can lead to cardiovascular diseases, (He &
MacGregor, 2011)), see Figure 10.

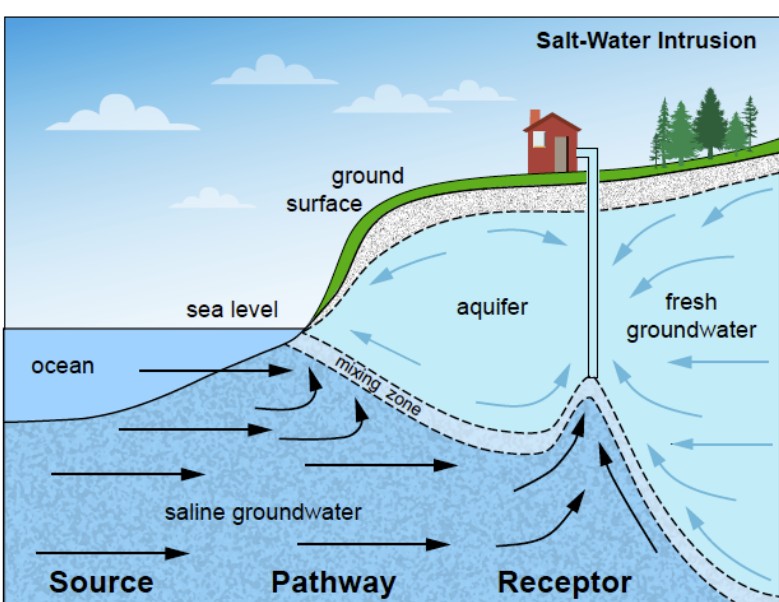

**Figure 10 A visualisation of the Source Pathway Receptor framing for saltwater impacts.**
Rivers and aquifers (good permeable water-bearing porous media) contaminated by high salinity decrease freshwater storage
and water quality, reduce soil fertility, impact on vegetation and freshwater species, and affect human health. Moreover, in
deltaic regions, SWI can also negatively impact ecosystem services and aquaculture activities such as clam or shrimp farming
(e.g., Dierberg and Kiattisimkul, 1996; Hou et al., 2022) (Figure 11). SWI is a slow but increasing hazard affecting European
coasts, especially in deltas and estuaries. In these low-elevation coastal zones, climatic changes (including sea-level rise)



combine with the changes induced by human activities such as reducing river flows causing salt-wedge shifts (Maselli &
Trincardi, 2013). Figure 11b shows several processes causing problems induced by SWI such as extracting groundwater,
creating controlled low-lying areas (polders) and reducing in urbanized areas fresh water depletion into the groundwater
systems (sealing). As a result of climate change (and associated SLR) and induced human activities, impacts of SWI are likely
to increase (e.g., (Befus et al., 2020; Oude Essink et al., 2010). In this section we discuss the main impacts of SWI, again using
the source (Section 6.2), pathway (Section 6.3), receptor/consequence (Section 6.4) framework.

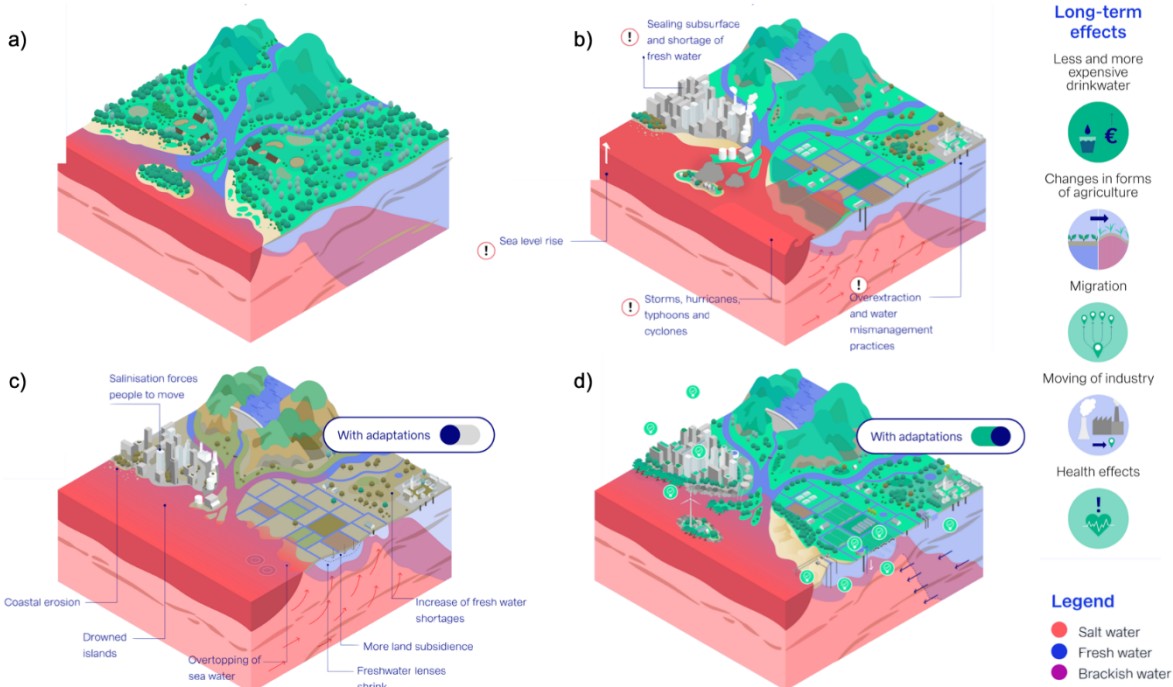

**Figure 11: Salinisation of (ground)water resources and land for different stages of progressive development. Salt water is
represented in red / pink, brackish water in purple, fresh water in blue. (a) Situation in the distant past, before human settlements;
(b) current situation; (c) situation in the future without adaptation; (d) situation in the future with adaptation. Source: Deltares
(2023).**

The scarcity of data often poses a challenge for sustainable management of groundwater resources worldwide. Mapping and
monitoring the spatial extent of near-coastal fresh groundwater resources usually requires detailed information for large coastal
regions, which is often not available whereas in-situ measurements are time- and labor consuming. To address this challenge,
remotely sensed data can be a cost-effective way to gather both surface and groundwater information, covering a large area in
a short period. For instance, Airborne Electromagnetic geophysical methods are particularly useful for detecting groundwater





salinity affecting the conductivity of the groundwater. Such methods have been executed in Denmark (Duque et al., 2022),
       Germany (Siemon et al., 2015) and The Netherlands (Delsman et al., 2018).

## 6.2 Source for surface saltwater intrusion

Surface SWI threatens water resourcing and freshwater availability especially in low-lying coastal areas such as deltas and
estuaries (van Engelen et al., 2022), which are characterized by natural complex interactions between fresh and salty waters
       (Horner-Devine et al., 2015; Valle-Levinson, 2010). The inland intrusion of salty waters along the river courses is controlled
       by the forces acting at both the river and the sea domains: the advective dispersion associated with the river flow, the steady
       shear dispersion associated with the estuarine exchange flow, and the tidal pumping (Lerczak et al., 2006). The balance among
       these forces is regulated mostly by the combined action of river discharge and sea level oscillations (Bellafiore et al., 2021).
Surface heat fluxes (evaporation) and the salt content in corresponding sea water also play a role in determining the extent of
       surface SWI. Therefore, while more evident in drought conditions (e.g., the extended 2022 drought in the Po Valley - Italy;
       Bonaldo et al., 2023), the processes regulating SWI, and the concurring effects cannot be attributed just to one single driver.
       Moreover, these natural drivers can act both on short timescales, as tidal fluctuations, storm surges and hurricanes, and at the
       long timescales, as climatic fluctuations, subsidence and, among others, SLR. In general, the variation of surface SWI does
not only affect the extension of the affected area, but can lead to the predominance of some hydrodynamic processes, as, for
       example, shifting the system from a diffusive to an advective dominated river dynamics (e.g., the Po River). Also, a
       modification of the environmental conditions in transitional areas can vary the extent of eu-, poly-, meso- and oligo-haline
       areas (Rodrigues et al., 2019). The concurring processes affecting surface SWI are linked to progressive river discharge
       decrease, increased surface heat fluxes (evaporation) and increase in the salt content and in the relative sea level. In Europe,
several estuarine and deltaic systems are suffering from the progressive increase in surface SWI. Not surprisingly, this process
       occurs both in micro and macro tidal environments (e.g., the Po River and Elbe Estuary, respectively), with different sea
       salinity values (ranging from less haline ones in the Atlantic or North Sea to the more haline ones in the Mediterranean) and
       higher or lower surface heat fluxes. These environmental conditions trigger the predominance of one of the several saltwater
       drivers. As an example, the Tagus Estuary is exposed to a tidal excursion of up to 3.8 m, also amplified by resonance. Therefore,
even in normal conditions, sea saltier water intrudes affecting 43% of the estuary area (intertidal zone). The combination of
       tides and the periodic exposure to droughts, with several low discharge events, seem to massively affect the system (Rodrigues
       et al., 2019).

       The Atlantic coasts already include cases of increased saltwater intrusion in the groundwater system (Figure 12). Studies on
       the quantification of effects of changed drivers show their relative effect. In the Minho and Lima estuaries, in the northern
coast of Portugal, the future SLR scenarios identify a progressive increase in saltwater extension, and the effect of most extreme
       SLR scenarios is leading to a transgression of the saltier front of several kilometers (Pereira et al., 2022). In this case, SLR is
       identified as the dominant driver for increased saltwater intrusion, compared to future river discharge reduction. On the other
       hand, the quantification of the relative effect of SLR and reduced river discharge in future climate change scenarios leads to





opposite conclusions in a microtidal Mediterranean system, as the Po River delta. River discharge reduction affects SWI more

than SLR (Bellafiore et al., 2021). SLR and climate change induced salinization is predicted to worsen also in several coastal

locations in the North Sea (e.g., The Netherlands - Bonte and Zwolsman, 2010; Belgium - Bertels and Willems, 2022). In all

deltaic and estuarine systems, the evaluation of changes in drivers should be carried out considering possible additional long-

term modifications of the morphologic environment, such as subsidence.

Added to these natural factors and often acting in combination with them, several anthropogenic activities can exacerbate SWI

by lowering the surface freshwater supply. Examples are changes in land use and land drainage, irrigation, hydropower

production and over-exploitation of coastal aquifers.

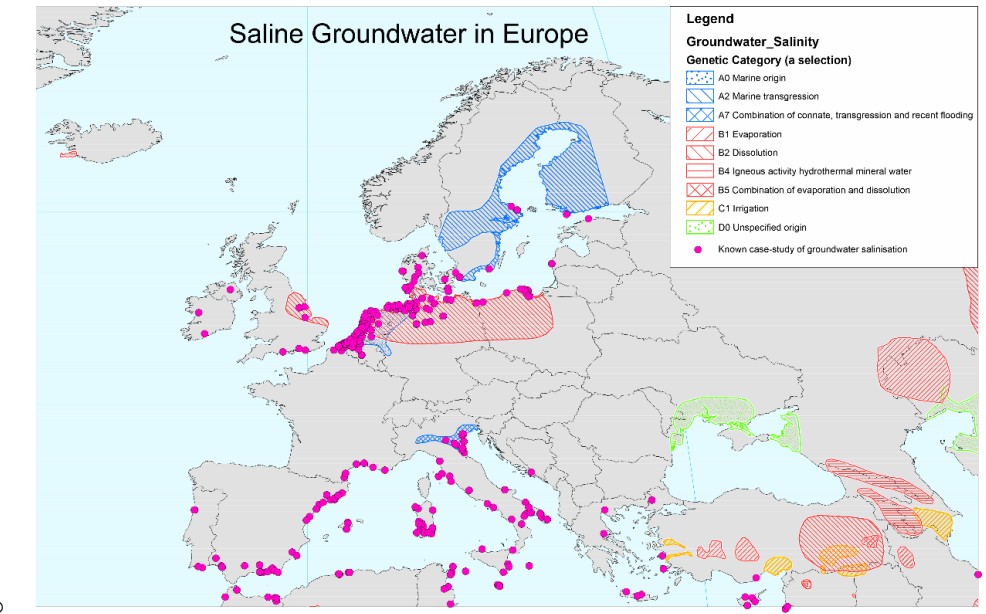

**Figure 12: Case-studies where groundwater salinisation is occurring in the coastal zone around Europe (updated and after Post et**

**al., 2018, and Van Weert et al., 2009), where it may cause multifaceted impacts combined with sea-level rise changes.**

### 6.3 Pathway for saltwater intrusion

For SWI, the pathway reflects how sea water reaches coastal freshwater resources causing their salinization. In estuaries and

deltas, rivers are the preferential way to transport sea water upstream, through salt-wedge intrusion along the riverbed and by

lateral inflow of the river into the aquifers. Land salinization and aquifer contamination can also result from coastal inundation

with saline waters, through floods induced by storm surges. The coastal morphology and geological characteristics are

therefore strongly influencing the saltwater pathway determining complex hydrogeological interactions between groundwater,

surface water and marine water. As explained in Section 6.2, climate change may enhance SWI drivers by raising the mean

relative sea level as well as by reducing the net freshwater supply in rivers and aquifers. However, as for coastal flooding, SLR

will also shorten the pathway of saltwater to reach land and freshwater resources by squeezing the coastline, thereby facilitating





coastal inundation and intrusion of marine waters. In addition, several anthropogenic activities exacerbate SWI by altering the river mouth and coastal morphology, such as diversions of water bodies, river channel deepening, saltmarsh reduction and human induced subsidence (White and Kaplan, 2017).

Several prevention and adaption measures have been undertaken in the last decades to limit coastal inundation and ingression of salty waters along the river channels and the aquifers in Europe. Anthropogenic interventions can affect SWI--impacted areas by increasing the downstream flow of freshwater (e.g., river diversion, optimization of freshwater withdrawals and deliveries) or by preventing the upstream transport of saline water. All flood barriers and measures limiting land inundation described in Section 4 have the co-benefit of limiting salinization of soil and intrusion of salt waters into surface and

groundwater systems. The most adopted engineered strategy to prevent salt wedges from intruding in estuaries and deltas is the installation of (often submerged) mechanical barriers (gates, dams, dikes, levees) near the river mouth that physically block the upstream flow of saline water (White and Kaplan, 2017). However, salt barriers are regularly damaged and breached during floods and are not effective during extreme droughts when the saline layer occupies the largest portion of the water column. Regarding groundwater, subsurface barrier walls (such as sheet piles, clay trenches and injection of chemicals) are considered

as one of the most effective methods for inhibiting SWI (Armanuos et al., 2020).  Various coastal Managed Aquifer Recharge (MAR) schemes can be implemented to mitigate groundwater salinization in the coastal zone (Dillon et al., 2019; Oude Essink, 2001) (see Figure 13). Sprenger et al. (2017) identify successful coastal MAR systems for The Netherlands (Amsterdam, The Hague areas), Spain (Barcelona Llobregat delta), Italy (Po delta), Belgium (De Panne), and Portugal (Algarve). These include increasing artificial recharge in upland areas to enlarge the outflow of fresh groundwater through the coastal aquifer, injecting

or infiltrating (purified) fresh water near the shoreline to create freshwater injection barriers, land reclamation to create a foreland where a freshwater body can develop or delay the inflow of saline groundwater. This is for instance a consequence of the sand suppletion ('The Sand Engine' at Zuid-Holland, The Netherlands) where the foreland better protects the low-lying hinterland against flooding (Huizer et al., 2016).

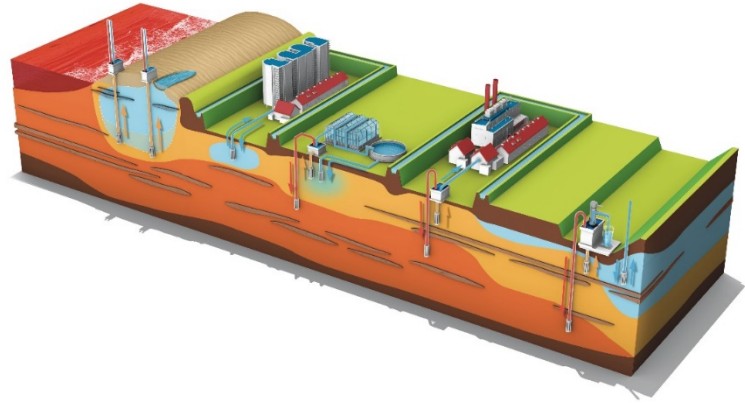




**Figure 13: Concept proposed for managing water resources in coastal areas, ensuring a reliable supply of freshwater for domestic, industrial, and agricultural purposes. To store excess freshwater, large-scale coastal Managed Aquifer Recharge (MAR) and Aquifer Storage and Recovery (ASR) methods can be used (see text). Additionally, brackish groundwater can be intercepted to prevent the salinization of aquifers and low-lying areas, while also providing a source of freshwater after desalination (after**
**https://www.coastar.nl/ COASTAR, 2018).**

### 6.4 Receptor and consequences of saltwater intrusion

The progressive salinization of water resources has severe and long-lasting consequences on several social (e.g., reduction of drinking water reservoirs), economic (e.g., water systems, agricultural production and land losses), and environmental (e.g.,
degradation or losses of freshwater habitats, changes in biodiversity) issues. According to Cooper et al. (1964), groundwater SWI is a natural occurrence in coastal groundwater systems. However, human activities such as excessive freshwater pumping from coastal aquifers (Custodio, 2002; Custodio & Bruggeman, 1987; Mastrocicco & Colombani, 2021; Schmork & Mercado, 1969) often disrupt this natural process and even cause land subsidence (Minderhoud et al., 2017). The interaction between groundwater and surface water can also be significant, with saline groundwater sometimes exfiltrating towards surface water
systems (De Louw et al., 2011; Delsman, 2015) and negatively impacting agricultural use and nature (Stofberg et al., 2015).

In densely populated and industrialized coastal regions around the world (Neumann et al., 2015b), including areas in Europe, groundwater often serves as the primary source of freshwater. Several studies have examined the future availability of groundwater worldwide under climate change and related SLR (Green et al., 2011; Taylor et al., 2013) and human activities (e.g., Wada et al., 2010, 2014). Projected SLR is expected to exacerbate water stress in these densely populated areas,
potentially leading to overexploitation and salinization of groundwater resources due to upcoming saline water (Figure 14. This situation is further compounded by the growing demand for freshwater in the future. Groundwater extraction in deltas leads to accelerated salinization of coastal freshwater aquifers and land subsidence if groundwater infiltration is limited by low-permeable layers (Herrera-García et al., 2021). The combination of land subsidence, SLR and sand mining in riverbeds also increases the risk of SWI in estuaries and floods (Eslami et al., 2021).

The influence of SLR on coastal groundwater systems has been studied since the 1990s (Navoy, 1991; Oude Essink, 1996; Sherif & Singh, 1999). More recent contributions concern (global) conceptual analyses based on analytical comparisons of fresh-salt interfaces (Chang et al., 2011; Chesnaux, 2015; Chesnaux et al., 2021; Ferguson and Gleeson, 2012; Mazi et al., 2013; Werner and Simmons, 2009), 2D cross-sectional model studies (Ketabchi & Jahangir, 2021; Michael et al., 2013), and 3D model studies (Befus et al., 2020; Loáiciga, 2009; Mabrouk et al., 2018; Masterson & Garabedian, 2007; Oude Essink et
al., 2010; Rasmussen et al., 2013; Vandenbohede et al., 2008; Delsman et al., 2023), whereas the effects of SLR on coastal groundwater systems has been reviewed in Ketabchi et al. (2016).



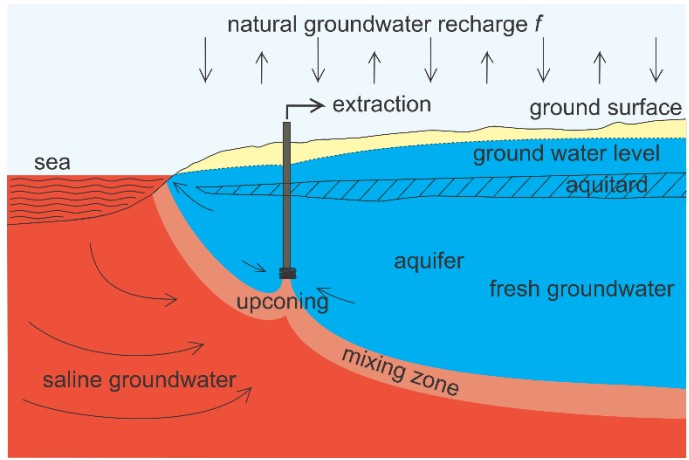

**Figure 14: Saltwater intrusion in a coastal aquifer where too much groundwater is extracted leading to upcoming of saline**
**groundwater into the extraction well.**

## 7 Ecosystems and estuaries

SLR can negatively impact coastal ecosystems and estuaries (e.g., degradation of marshes, wetlands, salt water intrusions), either through SLR itself or indirectly through measures implemented to protect the land from flooding or other adverse effects. On the other one hand, several coastal ecosystems can only be adapted or be maintained under narrow ranges of SLR rates
(e.g., (Timmerman et al., 2021). On the other hand, coastal defenses and measures implemented to protect coastal zones from SLR, such as flood barriers (e.g., Orton et al., 2023) also influence coastal ecosystems. These impacts can include increased water levels in estuaries, which may lead to the inundation of adjacent low-lying lands, shoreline erosion, and the potential failure of coastal and stormwater infrastructures (Hanslow et al., 2018; Pachauri et al., 2014; Sweet and Park, 2014). As such, the sustainable management of estuaries, adjacent low-lying areas, and their associated ecosystems requires a thorough
understanding of the complex impacts of (accelerating) SLR on estuarine processes (Saintilan et al., 2018; Schuerch et al., 2018). This knowledge will inform effective strategies for adaptation and mitigation in the face of SLR challenges.

Estuarine habitats, located at the interface between land and sea, face significant pressures from anthropogenic stressors originating within their catchment areas, including sedimentation, eutrophication, and pollution (Smith et al., 2013). These stressors often result from excessive inputs of terrestrial sediment, nutrients, and contaminants. Additionally, the estuaries'
ecosystems are vulnerable to the effects of global-scale climate change, including SLR and associated impacts. However, the ecological consequences of SLR and related factors have received relatively little attention (Beukerna, 2002; Rullens et al., 2022). SLR will have a significant impact on coastal and ecological risks, including the reduction and degradation of coastal biodiversity. While further data and analysis are needed to fully understand the combined evolution of these effects, it is evident that the decline in ecosystem services will exacerbate the vulnerability of coastal systems. Once these systems reach a





threshold or tipping point, their restoration becomes challenging, and their ability to adapt to new climate or management

conditions is severely limited.

Intensified flooding and erosion resulting from SLR pose increasing risks to coastal activities and infrastructure (Genua-Olmedo et al., 2016; Sections 4 and 5). These risks are often projected to amplify under future climate scenarios. A reduction in riverine fluxes contributes to a sedimentary deficit and accelerates relative SLR (Ericson et al., 2006). To restore such

systems effectively, progressively larger volumes of sediment are required to compensate for SLR and subsidence (see Sea Level Rise in Europe: observations and projections), maintaining the relative sea levels and ensuring deltaic sustainability.

**8 Conclusions**

In this chapter, the main impacts of SLR, namely coastal flooding, erosion and saltwater intrusion, have been reviewed using the concept of the Source-Pathway-Receptor-Consequence. Regarding coastal flooding, SLR, along with changes in storm

surge and wave height, and in some place's increases in tidal range, has driven an increase in the frequency with which extreme coastal water levels are exceeding high thresholds (Source). This in turn, along with an ongoing decline in the extent of natural habitats that act as a buffer to flooding (Pathways) and rapid population growth and urban encroachment in flood-prone areas (Receptors), has driven an increase in coastal flooding and its impact (Consequences) around the coast of Europe. However, current flood risk around the coastline of Europe would be considerably higher without the decades of investment into

extensive flood risk management infrastructure and advances in flood forecasting and emergency response. At the same time, losses in major events that exceed defence design standards are growing and are expecting to increase massively in the future with higher rates of SLR, unless further adaptation is taken. Furthermore, events with low flood levels (i.e., nuisance flooding) are likely to increase causing widespread disruption of everyday routine activities and property damages, especially in low lying areas.

Coastal erosion shares the same source as flooding; i.e. extreme waves and storm surge, even though in many areas additional long term anthropogenic, geological and climatic factors are also important, as they affect sediment budgets (Pathways). A total of more than 8,200 km of Europe's sandy beaches have significantly retreated over the last decades, found at various locations across the continent (Receptors). Climate change and rising seas are expected to accelerate the current trend and retreating shorelines combined with built areas backshore will result in coastal squeeze and threaten sandy beaches. The above

will not only have social, economic, and ecological consequences, but is expected to also exacerbate coastal flooding.

In many regions in Europe, saltwater intrusion into surface and groundwater systems is emerging as a problem. Sea-level rise is an important driver (Source). SLR facilitates coastal inundation and intrusion of marine waters into fresh water resources (Pathway). But other (human) processes also play an important role such as groundwater extractions and low-lying areas

attracting salt water and reduced fresh river flows.  These processes diminish fresh water availability leading to health risks caused by drinking too salty water, less fresh water for economic purposes (e.g. salt damage to crops) and nature (stresses on



existing biodiversity). A wide variety of adaptation measurements has been developed over the last decades, reducing the impact of saltwater intrusion. At the same time projected SLR is expected to increase the impact of saltwater intrusion in the densely populated coastal areas of Europe, particular in combination with the growing demand for freshwater in the future.

In summary, we can conclude that the impacts of SLR are emerging at many places across Europe and it might be expected that these impacts on freshwater availability will increase over time, thereby providing an incentive for further mitigation measures whereas at the same time smart adaptation measures are needed to reduce the impacts itself.

**Author Contributions**

RvdW and AM coordinated the paper. RvdW let the writing of the abstract. Section 2 was led by PL. IH and RvdW drafted section 3. IH and AT drafted section 4. MV, PC and AL drafted section 5. GUO, CF, DB drafted section 6 and JS section 7. All author contributed to the different versions of the entire manuscript. Kate Davis from Southampton University drafted the figures 2,3,7 and 10.

**Competing interests**

The contact author has declared that none of the authors has any competing interests.

**Acknowledgements**

A. T. acknowledges financial support from the Ministerio de Ciencia e Innovación (MCIN/AEI and
NextGenerationEU/PRTR) through the Ramon y Cajal Programme (RYC2021-030873-I). I.D.H time was supported by the UK Natural Environment Research Council grant CHANCE (NE/S010262/1). AM and AT were supported by CoCliCo, which has received funding from the European Union's Horizon 2020 research and innovation programme under grant agreement No 101003598. D.B. and C.F. contribution is embedded in the scientific work of the ESFRI DANUBIUS Research Infrastructure – The International Centre for Advanced Studies on River-Sea Systems (http://www.danubius-
ri.eu/). RvdW received funding from the NPP.





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
