# Peer review of "Sea Level Rise in Europe: impacts and consequences"

_State of the Planet, 2023_

## Referee Comment (RC2)

The paper "*Sea level Rise in Europe: impacts and consequences*" aims at providing an overview of the main type of impacts of SLR on the coast at the European scale. Three main processes are considered, i.e. flooding, erosion and saltwater intrusion, all having widespread impacts at the coast, also depending on the local setting. So it is not easy to have an overview at the continental scale. Moreover, the three processes are frequently overlapping, and this is addressed in the paper (such as in section 5.6). However, some repetition among chapters is present (see comments below) and might be avoided.

I indicate below, for each section, some comments. Additional minor suggestions are listed at the end of this document. Overall, I suggest minor revision.

Specific comments

The paper frequently refers to IPCC report and previous literature; in some cases, I found reported data not suitable to the case. This is the case of **section 2**, lines 129-130: "(…) *it is suggested that by 2100 coastal retreat could reach approximately 100 m for a 4°C temperature increase*". This overall estimate is not appropriate, since coastal retreat largely depends on coastal slope and beach morphology (without considering sediment supply but "pure" SLR) so it should not be indicated as a constant value for any degree of global warming.

Again in section 2, at line 131 the possible threat to UNESCO World Heritage Sites is mentioned but no references are provided. A couple of papers dealing with WHS at risk for coastal flooding and SLR are, in case:

Reimann et al., 2018. *Mediterranean UNESCO World Heritage at risk from coastal flooding and erosion due to sea-level rise*. Nat Commun 9, 4161. https://doi.org/10.1038/s41467-018-06645-9

Romagnoli et al. 2022. *Coastal Erosion and Flooding Threaten Low-Lying Coastal Tracts at Lipari (Aeolian Islands, Italy).* Remote Sens.14(13), 2960; https://doi.org/10.3390/rs14132960

In **section 3** the common approach applied for the analysis, namely the SPRC framework, is introduced. I expected a longer section than the present one (that is just 7 rows), citing other examples of coastal studies where this approach has been successfully applied, such as (for instance):

Villatoro et al. 2014. An approach to assess flooding and erosion risk for open beaches in a changing climate, Coastal Engineering, 87, 50-76, https://doi.org/10.1016/j.coastaleng.2013.11.009.

Also, the citation of "traditional exposure vulnerability approach" might be supported by some references.

In Figure 2, the sketch of should also report the term "Consequences" to be complete, otherwise delete it from the caption (as it is in figures 3, 7, 10).

**Section 4** on coastal flooding is probably the more complete. I give below just some comments/suggestion.

In "4.1 Source", among interannual SL variability (lines 185-187) it might be also cited the role of oceanographic processes at the basin/sub-basin scale. For the Mediterranean, these are recent references:

Menna et al., 2022. *Climatic, Decadal and Interannual variability in the Upper Layer of the Mediterranean Sea Using Remotely Sensed and In-Situ Data*. Remote Sens. 2022, 14(6), 1322; https://doi.org/10.3390/rs14061322.

Meli, M. et al. 2023. *Sea-level trend variability in the Mediterranean during the 1993–2019 period*. Frontiers in Marine Science, 10:1150488. https://doi.org/10.3389/fmars.2023.1150488

Some estimates referred to the text are not easily checked from figures. For instance:

-at lines 288-290 it is mentioned *"the probability of compound flooding (….) is projected to robustly increase by the end of the 21ᵗʰ century"*. Based on figure 4b I understand that this larger probability increase is represented by $\Delta$ T of 40-80 % (orange to red colors), is it correct? It might be useful to quantify also in the text to help readers.

In Subsection 4.4, at lines 383-385 some estimates are given in the text for the increase in flooded area due to events with different recurrence times. I found not easy to compare these values with figure 6.

Subsection 4.5 on "initiative to develop flood-related climate services in Europe" is useful. Dealing with Adaptation, possible overlaps with related chapter of the Assessment Report might be checked and, in case, cross-reference is suggested.

**Section 5** on coastal erosion from my point of view needs some clarification and possible reorganization due to some repetition.
The section mostly deals with processes affecting low-lying costs, i.e. the most threatened by SLR. A cliffed coast is instead represented in figure 7 (although erosion processes and impacts of SLR on this type of coast are not really accounted in the paper). I noted this discrepancy also in the Abstract (lines 28-29) where cliff failure is mentioned for coastal erosion - I suggest to simplify that sentence in "*Coastal erosion leads also to damage and....(…)*" and avoid mentioning cliffs.
Also, the concept of "negative coastal balance=erosion" given in "5.2 - Pathway for erosion" (line 500) is, in principle, correct for beaches. In cliffed costs, erosion and shoreline retreat are directly related to the balance between assailing force of waves and resisting force of the cliff material, when the former exceeds the latter. That is another reason to clarify with type of coasts the chapter is focusing on.
On the other side, at line 511, where it is mentioned "(….) *factors that affect the erodibility of the beach*" it might be added "*and the cliff*".

A definition of coastal erosion is provided at line 459, in subsection 5.1. Other definitions might be confusing or not univocal, such as the above cited (at line 500) that "*coastal erosion takes place when the sediment budget of a given area becomes negative*", for the reasons explained above. I suggest deleting this sentence since it only reflects part of the process, whose complexity is better explained in the text just below (lines 500-511). Another definition is given at line 514, in "Receptor and consequences of erosion" ("*Coastal erosion is the process by which the land is worn away and is submerged in water")*; this might be misleading. Submergence is also due to episodic flooding, not necessarily to coastal erosion. A further definition at lines 578-579 ("*Erosion is a physical phenomenon where sand is removed from the shoreface and deposited elsewhere, usually offshore"*) is also misleading because sediment transport and deposition are parts of cyclic processes modelling the emerged and submerged beach and should not be confused with (permanent) sediment loss that leads to erosion.

Among factors indicated as drivers of coastal erosion in subsections 5.1 and 5.2, some concepts and terms might be more precise and inclusive, trying at the same time to avoid repetitions.
For instance, instead than "*terrestrial sediment supply*" (line 471) the indication of "*sedimentary balance of the coastal stretch*" would be better used, because it includes also marine sediment supply, not only terrestrial one.

Factors influencing the sedimentary balance at the coast, both natural (climatic) and anthropogenic, should encompass a view to river catchment basins. This link should be better evidenced in a source-to-sink approach. An example of this is provided by references below:

Meli and Romagnoli, 2022. *Evidence and implications of hydrological and climatic change in the Reno and Lamone river basins over the last century and in related coastal area, Emilia-Romagna, Northern Italy.* Water, 14, 20650, doi: 10.3390/w14172650

Other natural factors with possible negative effects on the coastal budget are the occurrence of canyons' heads in the nearshore, that can subtract sediment from longshore drift. This potential effect could be mentioned, since this process is more frequent than commonly considered.
Among "human" factors (cited at line 507) it might be also mentioned the reduction of natural defence capability of the beach (its own resilience) due to alteration of natural coastal dynamics, and the stiffening of the coast caused by the construction of "hard" protection structures.
To conclude, subsections 5.1 and 5.2 are partially overlapping. The Authors might consider if there a way to avoid this.

Among the coastal monitoring programs mentioned in subsection 5.3 it might be included the long-term, regular monitoring carried out since the 1983 on the Emilia-Romagna coast (regional scale) on the emerged and submerged beach and on the shoreline (see public reports, https://www.arpae.it/it/temi-ambientali/mare, and adopted indicators on coastal erosion, https://webbook.arpae.it/erosione-costiera/index.html).
In Table 1 -Summary of the methods for monitoring, for what regards Drones, it might be added something like: "*Marine drones such autonomous surface vehicles (ASVs) are usefully applied for monitoring the nearshore in shallow water and for testing the effects of mitigation strategies against erosion*"
Ref: Stanghellini et al., 2022; https://doi.org/10.3390/rs14225901

In subsection 5.4 "Historical shoreline change" it should be mentioned that many coastal areas are artificially stabilized, otherwise estimate given for erosion in Europe should be much worser.

In subsection 5.5 "Future shoreline change due to SLR" (I suggest adding this specific to the title), in the sentence regarding Mediterranean beaches (lines 557-558), I do not understand how the beach narrowness should depend on slope. Coarse-grained beach with higher slope may be narrow (but they are less vulnerable to SLR), while low-gradient, sandy beaches may be narrow or large for different reasons... and will be more exposed to SLR to due to their reduced height. Furthermore, I question the projections reporting shoreline retreat (at lines 561-562): it should be specified (also in the caption of figure 9) that these estimates only represent effects due to SRL (according to a "bathtub approach" with respect to terrain model, I suppose), but they do not take into account the coastal morpho-dynamic evolution neither sedimentary budget, as it would be requested for estimating shoreline retreat due to erosion. These further aspects are well addressed in subsection 5.6, but not clarified here.

**Section 6** on saltwater intrusion follows the same organization of previous sections (SPRC framework) and provides also some indications on possible adaptation measures. Very minor corrections to text and figures are indicated below.

**Section 7**: it is not clear to me the reason for this section dealing with impacts of SLR on specific environments (estuaries) and ecosystems. The impacts include all the three previously considered

(flooding, erosion, saltwater intrusion) so it cannot be easily moved into one of the previous sections as it is. However, I found it disconnected to the rest of the paper.

The last paragraph (lines 793-796) is instead more general and partly overlaps with other sections of the chapter, also repeating some previous concepts; consider moving it or deleting it.

Minor comments and technical corrections:

-**lines 56, 97**: check parenthesis (too many open)
-**lines 113-114** there is a repetition of what already reported at lines 107-108.
-**line 298** in the sentence "(…) *can also erode riverbanks and cause landslides, leading to further flooding*" the Author would probably mean the "breaking of the embankments/levees".  This would be more correct than a general term "landslides".
-**line 323**: you could rewrite as "(….) *and stabilization of beaches and dunes*"
-**line 330-331**: "(….) *a continued decline in the extent of natural systems* (…)", this reduction in the extent can be very well expressed with the concept of "coastal squeeze".
-**line 467**: I would delete citation "Romagnoli et al., 2022" from here (it is also mentioned just below as local-scale study).
-**line 647**: in the caption of Fig. 11 it might be specified "(b) *current situation, with effects due to climate change and human activities*". Note that in the sketch c) the word "subsidence" should be corrected.
-**line 693**: "(…) *such as due to subsidence*" would sound better.
-**Figure 13**: this figure might be more extensively explained, it is very cryptic. SAR seems not mentioned in the text but only in the figure caption.
-**line 755**: a parenthesis is lacking at the end of the sentence "(*Figure 14*)".
-**Figure 14**: correct in the caption "*upconing*", it is the scientific term for the process indicated in figure.
-**line 774**: delete "other" from "*On the one hand*".

Claudia Romagnoli, 31/01/2024

---

## Author Comment (AC1)

This paper reviews impacts and consequences of sea-level rise (SLR) in Europe. This is not an easy task given the heterogeneity of the literature on this topic, so the authors should be congratulated for their effort.

Thank you

To do so, the authors have adopted a focus on the physical processes causing flooding erosion and salinization, rather than e.g. a focus on different types of coastal areas in Europe. The advantage is that it gives a useful reminder of the physical mechanism, the drawback is that it takes always some time before each section start addressing issues specific to Europe.

Besides this focus on impacts, some information is provided on consequences, e.g. on erosion based on previous publications, but overall the information on consequences is limited. To me this is not a surprise given the heterogeneity of information available on the consequences of SLR in Europe and the uncertainties, but may be it would be useful that the authors clearly say that we are lacking aggregated observations of consequences and – if they agree with me – the information on projections based on modelling are often difficult to assess and compare due to model assumptions and limitations.

We have tried to sharpen the purpose of the paper in the second paragraph and included this comment.

The terminology used in the paper is clear overall, except sometimes when it comes to the terminology on hazards and adverse events (see ref below to the UNISDR and IPCC terminology). This should be clarified.

We aim to streamline the terminology by using the concept of source, pathway and receptor which we introduce in a separate paragraph and consequently use throught the remaining text.

The main problem of the paper, from my perspective is a lack of clarity on its objectives : who is the target audience? What will they learn in this article? This is not quite clear in the current version and could be precised at the end of the introduction and reflected in the conclusion.

This paper aims to describe the different impacts of sea-level rise in Europe following from the physical evidence expressed by Melet et al. (2024) and is intended for local and

governmental stakeholders planning for raising awareness and consider adaptation measures in their region. We have put this upfront in the paper.

Below are specific comments

Specific comments (moderate)

1. Introduction

The introduction is interesting, but it would benefit from a bit of reorganisation and update. For example, the statements on exposure increase rely on relatively old references published in 2003. Meanwhile, as recognized by the authors line 65, setback zones have been setup, and this may have curbed trends in exposure, at least in some contexts. Furthermore, it would be good to remind that Europe has a long history of coastal protection compared to many other regions in the world already at this stage of the manuscript. Finally it is not clear whether the whole introduction is focused on present days, or both present days and the future. In the first case, it would be good to cover the topic of attribution of coastal impacts to sea-level rise. In the second case it would be good to separate clearly what applies to today, and what is projected to 2050, 2100 or beyond. For example migrations patterns could be different today and in 2100 when sea-level rise becomes a major driver of coastal migration.

One way to clarify this could be to reorganize the introduction in a way that explains more clearly what is known with "high confidence" (e.g., flood risk will increase) from what is uncertain (e.g. the precise magnitude of flood or erosion impacts) or almost unknown (e.g., future migration responses to SLR). Another option would be to remind the target audience of the paper, their potential challenges and how this paper is going to help them. There may be other options.

Added a sentence arguing that coastal migration is more in the future and hard to quantify.

Added a sentence on the high confidence

2. Summary of previous assessments

The review makes sense overall, and it is good to remind where coastal impacts can be found in AR6, but it is unclear to me how useful it is for a review focused on Europe: in practice, the authors could well say in the introduction that there has been previous assessments on climate change in Europe (IPCC), on specific coastal hazards (Eurosion…), on some relevant regions (Medecc), but that the Knowledge Hub is the first attempt to do what it does. So in summary I am not sure such a long summary is needed, may be a few sentences in the introduction would be sufficient.

Besides, there are several points that could require some precisions (see below, minor comments)

We believe that the strength of the section is not only that it points out where the information is in the AR6 reports but that it ditails the information which is specific for Europe. To clarify this, we changed the title of the section to express this rather than incorporating the information in the introduction, which would then get too long.

3. Source, Pathways, Receptors and consequences framework

The Source, Pathways, Receptors and Consequences framework section is very short. I wonder if it could be included in the introduction as this is framing the rest of the manuscript.

We agree this section is short, however we deliberately keep this section short, because it is simply introducing the concept, which is then explored in depth in the sections below. We have added an additional sentence, on the advice of the other reviewer to gives some examples of where it has been used previously.

4. Coastal flooding and compounding flood events

This section has a focus on physical processes causing flooding. It is good to have a section on compound flooding, but it may be not clear to readers how increased precipitations due to climate change (higher atmospheric moisture due to increased temperatures) have been considered in the references cited.

This is a really good point, thank you. We have added the following sentence to stress this:

*'However, in regard to compound flooding, climate change also impacts precipitation (as average temperatures increase, more evaporation occurs, which, in turn, increases overall precipitation) and therefore river flow.'*

In the sentences below, we do reference some papers regarding changes in precipitation.

The discussion on pathways highlights the need to maintain defenses in order to maintain security levels, which is good. However, it does not mention the risk of coastal defenses being raised at the costs of coastal biodiversity losses (see e.g. Bednar-Friedl et al., 2022 p 1843).

Thank you – this is a good point. We have added the following sentence citing that suggested paper:

*'There is also concern that raising existing coastal defences, or building new ones, will come at the cost of further biodiversity losses (Bednar-Friedl et al., 2022).'*

Another point which is missing is the projected change in terms of flooding modes: from overtopping to overflow, as shown in IPCC AR6 WGII CCP4 page 2245. This risk is important as areas currently exposed only to overtopping may not be prepared to future overflow and the associated risks to human life and larger economic damages. These aspects should at least be mentioned for completeness.

Again, this is a very good point and we have added the following sentence, and your suggested reference:

*'With higher SLR coastal flooding will progressively change, due to changes in the pathway from overtopping to overflow, high-tide flooding and ultimately permanent flooding (Ali et al., 2022).'*

5. Coastal erosion

It is good (and not a surprise given the list of authors!) that the section identifies the topic of erosion/flooding interactions. There is also a summary of observed and projected erosion, which can be useful. However, it should be precised that these figures assume no geological constraints (to be confirmed by the authors – see Thiéblemont et al 2019 for this caveat) and no protection or nourishment.

The reviewer is raising an important point, especially for the projected values (the baseline ones come from observations and are the result of different meteorological, geological and antropogenic factors). Therefore, we have added a small section discussing geological constraints, citing relavant papers including the one suggested by the reviewer.

This section is structured in a different way than the other ones on coastal flooding and salinization, which is not necessarily a problem, but raises the following questions: (1) why a section on monitoring for erosion and not flooding and salinizations? Why not an analysis on historical and future flooding and/or salinization?

It is true that this may seem not conistent but we believe that historic shoreline change trends are a key component for projecting the future shoreline positions. As such, understanding more about the (un)certainty in such historic trends is important when projecting into the future. This is different in case of flooding and salinization.

6. Saltwater intrusion

I am less qualified on this topic and may not cover all subsections adequately. In general the section recognizes well the importance of aquifer recharge and water usages in driving current salinization in aquifers, but since the topic of this paper is on sea-level rise, a specific discussion on the impacts of sea-level rise or the vulnerability of different types of European aquifers could be useful. The Jouzel report in 2015 (in French) could provide some useful examples:
https://www.ecologie.gouv.fr/sites/default/files/ONERC_Climat_France_XXI_Volume_5_V F_revisee_27fevrier2015.pdf

We provided more references in the saltwater intrusion section

Figure 12 seems good to me, but it is not well exploited in the text. Furthermore it would be useful to say clearly that the known case study are not exhaustive - e.g. salinization in the Loire estuary has been reported during the 2022 drought with concerns to drinking water supply, and is not reported here.

Added "not exhaustive" to the figure caption.

Section 6.2 is clear, but I do not understand the focus on river salinization – the statements on reduced recharge apply to aquifers as well, do not they?

Removed "surface" from the title.

In section 6.4, I am not sure that all messages apply to Europe (see detailed comments and the comment on sand mining in river beds).

The sentence referring to sand mining has been removed.

Finally may be one study to consider: https://doi.org/10.1029/2023EF003581

Reference added.

7. Ecosystems and estuaries

The good point in this section is the reminder that coastal protection is often implemented at the cost of coastal ecosystem losses, as already reminded by the IPCC WGII report. That said, this section does not fit very well in the structure of this manuscript: the title suggests it may either focus on a particular type of receptor (ecosystems), or a particular type of geomorphological features (estuaries). In both cases it is not clear why this receptor or type of environment is not addressed in the previous sections. Besides, if a focus on ecosystems could be useful if it could give insight on how healthy ecosystems can reduce the hazards above, but this is not addressed here. I suggest reconsidering this section.

We agree and decided to remove this section as it did not fit.

8. Conclusion

The conclusion could include a summary of gaps of knowledge / recommendations for future assessments.

We added that impact of coastal ecosystems is one of the major knowledge gaps at the moment.

Minor comments

- line 41: suggest: these threats "can be" reinforced by subsidence caused by human activities

Changed accordingly

- line 45: consider replacing "losses" by "risks"

Changed accordingly

- line 47: Vousdoukas et al 2018b focus on extreme water levels. Therefore the statement on trends on exposure can not come from this paper. Suggest to either refer here to papers assessing impacts for various SSPs or to delete "with the relative importance of trends in exposure (related to coastward migration, urbanization and rising asset values) diminishing over time", as the next sentence is about the same topic.

Sorry we referenced to the wrong paper. We also rephrased slightly to clarify

- line 49: The discussion on exposure in the introduction relies on papers published a long time ago (20 years). Meanwhile, setback zones have been defined in many countries to reduce exposure (e.g. Croatia, France...). This may have had an impact on exposure trends or not depending on the context. It would be good if the authors could either find a paper addressing this issue, or identify a gap of knowledge.

We have included this in the conclusion section at the end.

- line 98 and earlier: the SYR builds on all AR6 reports, not only SROCC as suggested.

Corrected

- line 116: please note this is expected annual damage, not annual damages. Furthermore these are projections based on flood models with coarse resolution and strong assumptions must be made on protection and exposure. Therefore these figures are indicative.

Rephrased

- Line 125: in Mediterranean ports the problem is also that waves are projected to propagate within ports with shallow bathymetry (see page 2244 in IPCC AR6 CCP4)

Rephrased

Line 134: It is true that one of the four key risks of Chapter 13 is the impacts of heat to ecosystems and that the marine heatwaves are causing massive mortalities, especially in the Mediterranean, as reported in CCP4, so the statement is certainly valid for coastal aquatic ecosystems. Yet in practice for coastal ecosystems another risk that is mentioned several time in the report is a massive use of hard engineering protection in response to coastal hazards and SLR, and therefore reducing habitats of coastal ecosystems (see page 1843 in Chapter 13). Coastal squeeze, not necessarily related to SLR, is also a major threat. To contextualize, references to IPBES here would be useful to remind the 5 major causes of ecosystem losses, which all apply to coastal ecosystems.

Rephrased

Line 144: Hazard is a potential occurrence of an adverse event (see https://www.ipcc.ch/report/ar6/wg2/downloads/report/IPCC_AR6_WGII_Annex-II.pdf) here the source describes the origin of the event (not the hazard)

Line 145: the "entity" could be named exposed element
https://www.preventionweb.net/files/7817_UNISDRTerminologyEnglish.pdf

We have reworded so the Source is described as the origin of the event that cases flooding. Entity simply means the thing that is flooded. We have also replaced entity with exposed element. Thanks for your suggestions.

Line 168 – Fig 3 – the figures seems to assume overtopping only but there are other modes of flooding to consider (overflow, breaching)

We have rephrased the caption and mentioned the other two examples

Line 171 – "governing hazards" –confusion between adverse events and hazards (see above comment line 144)?

We have replaced governing hazard with event.

Line 209 – the fact that lower tidal range mean higher sensitivity to SLR is clear but why this precise threshold of 2m here?

Adjusted

Line 337 – some figures are available on www.eurosion.org (2004)

We imply in peer reviewed papers

Line 342 – reference needed.

We removed this sentence

Line 463 – the sentence is not clear to me (?)

We have rephrased the paragraph, and we are confident that it reads better now.

Line 498 – Figure 7 – on this figure it is not very clear what the pathway is.

With coastal erosion it is hard to clearly show the pathway in an illustration. However, it is the pathway by which water gets to the base of the cliff, and so is impaced by the state of the tide, width and slope of the beach. Also, if there is a big cliff fall, sediment can lie on the beach for many months, preventing further erosion, as illustrated in Figure 3.7.

Line 534 – "Earth Observation" may refer to satellite and in-situ monitoring – suggest to change to Earth observation from space or similar.

Done

Line 545 and around: important to mention that erosion in www.eurosion.org can be assigned to coastal cliffs retreating at about 20cm a year in average, well above the detection threshold of Luijendijk et al 2018 – therefore the two datasets can not be compared.

Here, we are not comparing the two studies on erosion rates. The comparison is only made on the occurrence of sandy beaches (in %); see lines 552 "Analysis on satellite-derived sandy beach detection reveals that about 35% of the European coastline is sandy, which agrees largely with the 40% Eurosion estimate (EUROSION, 2004)." We agree with the reviewer that comparing erosion rates would not be appropriate given the accuracy of SDS, coastal types included, and different time frame.

Line 554 – Figure 8 –this type of representation has the problem that points are superimposed, so it is very difficult to visualize anything. Furthermore the precision of the dataset is low (0.5m/yr) – in some areas e.g. upper Normandy – rates of 0.2m/year are causing trouble for land use planning and human security. One way to simplify the figure and give justice to the dataset would be to plot only hotspots, that is areas with large erosion rates (e.g. larger than 1m/year)

Thank you for the suggestion, we agree and we improved the figure such that is show the hotspots only.

Line 569 – I think this assumes no geological constraint – see Thieblemont et al 2019

We have added some lines on the geological constraints and control from 580-585 addressing this remark.

Line 574 – this is without protection and without consideration of geological constraints (unerodible layers beneath the beach) – TBC

We have added some lines on the geological constraints and control from 580-585 addressing this remark.

Line 632 – Figure 10 – suggest to add arrows to define precisely where is the source, the pathway and the receptor in this figure – note also that this assumes a porous, homogeneous and unconfined aquifer (e.g. not the case in many aquifers in southern france for example). Figure 11 shows some more complexity.

These figures are meant just as an illustration. We would prefer to leave it as it is.

Line 758 – The study is in Vietnam. In Europe river bed sand mining is quite strictly regulated by e.g. the water framework directive, isn't it? Is there any message applicable to Europe here?

The sentence has been removed.

Line 770 – Figure 10 and Figure 14 are quite similar – consider merging

Commented [CF1]: I do agree.

We agree and removed figure 3.14.

Line 776 – it would be good to cite work in Europe as it exists (see references in IPCC reports, e.g. Cooper et al., 2016…)

We added several references in the paragraph.

I hope this review is useful

Gonéri Le Cozannet, BRGM, 29/01/2024

Thanks Gonéri the answer is YES

Bednar-Friedl, B., R. Biesbroek, D.N. Schmidt, P. Alexander, K.Y. Børsheim, J. Carnicer, E. Georgopoulou, M. Haasnoot, G. Le Cozannet, P. Lionello, O. Lipka, C. Möllmann, V. Muccione, T. Mustonen, D. Piepenburg, and L.Whitmarsh, 2022: Europe. In: Climate Change 2022: Impacts, Adaptation and Vulnerability. Contribution of Working Group II to the Sixth Assessment Report of the Intergovernmental Panel on Climate Change [H.-O. Pörtner, D.C. Roberts, M. Tignor, E.S. Poloczanska, K. Mintenbeck, A. Alegría, M. Craig, S. Langsdorf, S. Löschke, V. Möller, A. Okem, B. Rama (eds.)]. Cambridge University Press, Cambridge, UK and New York, NY, USA, pp. 1817–1927, doi:10.1017/9781009325844.015.

Reimann L, Vafeidis AT, Honsel LE. Population development as a driver of coastal risk: Current trends and future pathways *ridge Prisms: Coastal Futures*. 2023;1:e14. doi:10.1017/cft.2023.3

Thiéblemont, R., Le Cozannet, G., Toimil, A., Meyssignac, B. and Losada, I.J., 2019. Likely and high-end impacts of regional sea-level rise on the shoreline change of European sandy coasts under a high greenhouse gas emissions scenario. *Water*, *11*(12), p.2607.

---

## Author Comment (AC2)

Rebuttal

The paper "*Sea level Rise in Europe: impacts and consequences*" aims at providing an overview of the main type of impacts of SLR on the coast at the European scale. Three main processes are considered, i.e. flooding, erosion and saltwater intrusion, all having widespread impacts at the coast, also depending on the local setting. So it is not easy to have an overview at the continental scale. Moreover, the three processes are frequently overlapping, and this is addressed in the paper (such as in section 5.6). However, some repetition among chapters is present (see comments below) and might be avoided.

I indicate below, for each section, some comments. Additional minor suggestions are listed at the end of this document. Overall, I suggest minor revision.

Specific comments

The paper frequently refers to IPCC report and previous literature; in some cases, I found reported data not suitable to the case. This is the case of **section 2**, lines 129-130: "(…) *it is suggested that by 2100 coastal retreat could reach approximately 100 m for a 4°C temperature increase*". This overall estimate is not appropriate, since coastal retreat largely depends on coastal slope and beach morphology (without considering sediment supply but "pure" SLR) so it should not be indicated as a constant value for any degree of global warming.

We removed the too detailed estimate.

Again in section 2, at line 131 the possible threat to UNESCO World Heritage Sites is mentioned but no references are provided. A couple of papers dealing with WHS at risk for coastal flooding and SLR are, in case:

Reimann et al., 2018. *Mediterranean UNESCO World Heritage at risk from coastal flooding and erosion due to sea-level rise*. Nat Commun 9, 4161. https://doi.org/10.1038/s41467-018-06645-9

Romagnoli et al. 2022. *Coastal Erosion and Flooding Threaten Low-Lying Coastal Tracts at Lipari (Aeolian Islands, Italy)*. Remote Sens.14(13), 2960; https://doi.org/10.3390/rs14132960

References are added. We have also included this additional reference, with directly address the impact of sea level rise and coastal erosion on UNESCO sites:

Haigh, I.D., Dornbusch, U., Brown, J., Lyddon, C., Nicholls, R.J., Penning-Roswell, E. and Sayers, P. Climate change impacts on coastal flooding relevant to the UK and Ireland. MCCIP Science Review 2022, 18pp. doi: 10.14465/2022.reu02.cfl, 2022.

In **section 3** the common approach applied for the analysis, namely the SPRC framework, is introduced. I expected a longer section than the present one (that is just 7 rows), citing other examples of coastal studies where this approach has been successfully applied, such as (for instance):

Villatoro et al. 2014. An approach to assess flooding and erosion risk for open beaches in a changing climate, Coastal Engineering, 87, 50-76, https://doi.org/10.1016/j.coastaleng.2013.11.009.

We deliberately keep this section short, because it is simply introducing the concept, which is then explored in depth in the sections below. Thanks for your suggestion regarding adding additional references. We have added the reference you recommend, and three other studies that use this framework:

Haigh, I. D., Nicholls, R. J., Penning-Rowsell, E. C., and Sayers, P.: Climate change impacts on coastal flooding relevant to the UK and Ireland, MCCIP Rolling Evidence Updates, 18 pages, https://doi.org/10.14465/2022.REU02.CFL, 2022.

Thorne, C.R., Evans, E.P. & Penning-Rowsell, E.C., 2007. Future flooding and coastal erosion risks, Thomas Telford Services Ltd. London, UK

Donovan, B., Horsburgh, K., Ball, T. and Westbrook, G. Impacts of climate change on coastal flooding. MCCIP Science Review 2013, 211-218, doi:10.14465/2013.arc22.211-218, 2013.

Also, the citation of "traditional exposure vulnerability approach" might be supported by some references.

We have referenced the following paper:

Nicholls, R., et al. (2008), "Ranking Port Cities with High Exposure and Vulnerability to Climate Extremes: Exposure Estimates", OECD Environment Working Papers, No. 1, OECD Publishing, Paris, https://doi.org/10.1787/011766488208.

In Figure 2, the sketch of should also report the term "Consequences" to be complete, otherwise delete it from the caption (as it is in figures 3, 7, 10).

Thanks for your comment, this is a good point. Receptors and consequences are essentially the same thing and different studies use SPR and SPRC interchangeable. As consequences are not reffered to in the figures, we have as you suggested removed it. Hence, we just refer to Source-Pathway-Receptor throughout.

**Section 4** on coastal flooding is probably the more complete. I give below just some comments/suggestion.

In "4.1 Source", among interannual SL variability (lines 185-187) it might be also cited the role of oceanographic processes at the basin/sub-basin scale. For the Mediterranean, these are recent references:

Menna et al., 2022. *Climatic, Decadal and Interannual variability in the Upper Layer of the Mediterranean Sea Using Remotely Sensed and In-Situ Data*. Remote Sens. 2022, 14(6), 1322; https://doi.org/10.3390/rs14061322.

Meli, M. et al. 2023. *Sea-level trend variability in the Mediterranean during the 1993–2019 period*. Frontiers in Marine Science, 10:1150488. https://doi.org/10.3389/fmars.2023.1150488

Added accordingly

Some estimates referred to the text are not easily checked from figures. For instance:

-at lines 288-290 it is mentioned "*the probability of compound flooding (….) is projected to robustly increase by the end of the 21th century*". Based on figure 4b I understand that this larger probability increase is represented by D T of 40-80 % (orange to red colors), is it correct? It might be useful to quantify also in the text to help readers.

Added the 40-80% in the text

In Subsection 4.4, at lines 383-385 some estimates are given in the text for the increase in flooded area due to events with different recurrence times. I found not easy to compare these values with figure 3.6.

We agree it is a bit confusing that we refer to the area in percentage, yet Figure 3.6 shows thea rea in Km$^2$. Because the results are presented in Paprotny et al. (2019) by Country it is not easy to summaries the results here in a single sentence in km$^2$. However, to help clarify we have added the following sentence referring readers to Table 4 in that paper:

*'for size of area flooding in km$^2$ see Table 4 in Paprotny et al. (2019) for more details.'*

Subsection 4.5 on "initiative to develop flood-related climate services in Europe" is useful. Dealing with Adaptation, possible overlaps with related chapter of the Assessment Report might be checked and, in case, cross-reference is suggested.

Cross reference added

**Section 5** on coastal erosion from my point of view needs some clarification and possible reorganization due to some repetition.

The section mostly deals with processes affecting low-lying costs, i.e. the most threatened by SLR. A cliffed coast is instead represented in figure 7 (although erosion processes and impacts of SLR on this type of coast are not really accounted in the paper). I noted this discrepancy also in the Abstract (lines 28-29) where cliff failure is mentioned for coastal erosion - I suggest to simplify that sentence in "*Coastal erosion leads also to damage and....(…)*" and avoid mentioning cliffs.

Adjusted in abstract

Also, the concept of "negative coastal balance=erosion" given in "5.2 - Pathway for erosion" (line 500) is, in principle, correct for beaches. In cliffed costs, erosion and shoreline retreat are directly related to the balance between assailing force of waves and resisting force of the

cliff material, when the former exceeds the latter. That is another reason to clarify with type of coasts the chapter is focusing on.

We agree with the reviewer mainly in the sense that at hard cliffed coasts negative sediment budget will result also in wave cut notches, and eventual cliff collapse and erosion, but in much larger time scales. We have added a phrase 'especially at sandy beaches' and we thank the reviewer for the suggestion

On the other side, at line 511, where it is mentioned "(….) *factors that affect the erodibility of the beach*" it might be added "*and the cliff*".

We have replaced the word beach with coast.

A definition of coastal erosion is provided at line 459, in subsection 5.1. Other definitions might be confusing or not univocal, such as the above cited (at line 500) that "*coastal erosion takes place when the sediment budget of a given area becomes negative*", for the reasons explained above. I suggest deleting this sentence since it only reflects part of the process, whose complexity is better explained in the text just below (lines 500-511).

We have changed the sentence to 'Sandy beach erosion takes place when the sediment budget of a given area becomes negative'.

Another definition is given at line 514, in "Receptor and consequences of erosion" ("*Coastal erosion is the process by which the land is worn away and is submerged in water*)"; this might be misleading. Submergence is also due to episodic flooding, not necessarily to coastal erosion. A further definition at lines 578-579 ("*Erosion is a physical phenomenon where sand is removed from the shoreface and deposited elsewhere, usually offshore*") is also misleading because sediment transport and deposition are parts of cyclic processes modelling the emerged and submerged beach and should not be confused with (permanent) sediment loss that leads to erosion.

We have added the word permanent in the sentence: 'Coastal erosion is the process by which the land is worn away and is permanently submerged in water'.

Among factors indicated as drivers of coastal erosion in subsections 5.1 and 5.2, some concepts and terms might be more precise and inclusive, trying at the same time to avoid repetitions.

For instance, instead than "*terrestrial sediment supply*" (line 471) the indication of "*sedimentary balance of the coastal stretch*" would be better used, because it includes also marine sediment supply, not only terrestrial one.

We are a bit confused with this comment as the list of factors is meant not to be exhaustive. All the factors we mention affect the sediment budget and terrestrial supply is one of them; while the other factors affect onshore transport, among others. So we don't think there is a false statement here.

Factors influencing the sedimentary balance at the coast, both natural (climatic) and anthropogenic, should encompass a view to river catchment basins. This link should be better evidenced in a source-to-sink approach. An example of this is provided by references below:

Meli and Romagnoli, 2022. *Evidence and implications of hydrological and climatic change in the Reno and Lamone river basins over the last century and in related coastal area, Emilia-Romagna, Northern Italy.* Water, 14, 20650, doi: 10.3390/w14172650

Other natural factors with possible negative effects on the coastal budget are the occurrence of canyons' heads in the nearshore, that can subtract sediment from longshore drift. This potential effect could be mentioned, since this process is more frequent than commonly considered.

We would like to thank the reviewer for the reference which we cite in the paper. We also mention the effect of canyons as a sediment sink and a source of erosion which is a very good point as well.

Among "human" factors (cited at line 507) it might be also mentioned the reduction of natural defence capability of the beach (its own resilience) due to alteration of natural coastal dynamics, and the stiffening of the coast caused by the construction of "hard" protection structures.

We are confident that this point is already touched in this section; e.g.:

'In principle, coastal erosion can be the result of any process that alters the sediment transport patterns. This can be either hydrodynamic (e.g., changes in wave intensity or direction, sea level, etc) (Sierra and Casas-Prat, 2014), related to the presence of obstacles like hard

structures (Loureiro et al., 2012; Noble, 1978), or factors that affect the erodibility of the coast (Feagin et al., 2019). '

To conclude, subsections 5.1 and 5.2 are partially overlapping. The Authors might consider if there a way to avoid this.

We understand the point and we have checked carefully in the manuscript. For homogeneity reasons we present all impacts using the Source, Pathway, Receptor framework. Before there is a general introduction with definitions and drivers so some overlapping is inevitable, but we don't think the same points are repeated. We have already made several changes and if the reviewer has further specific suggestions, we are happy to consider them.

Among the coastal monitoring programs mentioned in subsection 5.3 it might be included the long-term, regular monitoring carried out since the 1983 on the Emilia-Romagna coast (regional scale) on the emerged and submerged beach and on the shoreline (see public reports, https://www.arpae.it/it/temi-ambientali/mare, and adopted indicators on coastal erosion, https://webbook.arpae.it/erosione-costiera/index.html).

We thank the reviewer for the constructive comment and we have added the information in the revision.

In Table 1 -Summary of the methods for monitoring, for what regards Drones, it might be added something like: "*Marine drones such autonomous surface vehicles (ASVs) are usefully applied for monitoring the nearshore in shallow water and for testing the effects of mitigation strategies against erosion*"Ref: Stanghellini et al., 2022; https://doi.org/10.3390/rs14225901

We have added the reference in the Drones section where we now mention also terrestrial and floating cases of autonomous surveying.

In subsection 5.4 "Historical shoreline change" it should be mentioned that many coastal areas are artificially stabilized, otherwise estimate given for erosion in Europe should be much worser.

It is a fair point and we have added a relevant statement: "It is important to highlight that several of the accreting or stabilizing trends found in Europe are due to human interventions, either through beach hardening or nourishment projects."

In subsection 5.5 "Future shoreline change due to SLR" (I suggest adding this specific to the title), in the sentence regarding Mediterranean beaches (lines 557-558), I do not understand how the beach narrowness should depend on slope. Coarse-grained beach with higher slope may be narrow (but they are less vulnerable to SLR), while low-gradient, sandy beaches may be narrow or large for different reasons... and will be more exposed to SLR to due to their reduced height.

The reviewer is right and the statement was left there by mistake after edits by different co-authors. Beach narrowness may correlate with slope, but this is irrelevant to the context of the paragraph. We have rephrased the sentence to: "Mediterranean beaches are more susceptible to the negative effects of SLR because they are narrower as a consequence of the lower tidal range and milder wave climate"

Furthermore, I question the projections reporting shoreline retreat (at lines 561-562): it should be specified (also in the caption of figure 9) that these estimates only represent effects due to SRL (according to a "bathtub approach" with respect to terrain model, I suppose), but they do not take into account the coastal morpho-dynamic evolution neither sedimentary budget, as it would be requested for estimating shoreline retreat due to erosion. These further aspects are well addressed in subsection 5.6, but not clarified here.

De Santiago is not applying a bathtub approach but morphodynamic models which simulate shoreline change by including cross-shore and alongshore sediment transport formulations, short- and long-term processes and considering hard structures. Other cited works apply the Bruun rule considering beach slope data and to some extent sediment budgets. Of course, any empirical or process-based model has its limitations, but we think that it's beyond the scope of the current manuscript to go into such discussions.

**Section 6** on saltwater intrusion follows the same organization of previous sections (SPRC framework) and provides also some indications on possible adaptation measures. Very minor corrections to text and figures are indicated below.

Done

**Section 7**: it is not clear to me the reason for this section dealing with impacts of SLR on specific environments (estuaries) and ecosystems. The impacts include all the three previously considered (flooding, erosion, saltwater intrusion) so it cannot be easily moved into one of the previous sections as it is. However, I found it disconnected to the rest of the paper.

The last paragraph (lines 793-796) is instead more general and partly overlaps with other sections of the chapter, also repeating some previous concepts; consider moving it or deleting it.

We removed it.

Minor comments and technical corrections:

-**lines 56, 97**: check parenthesis (too many open)

Corrected

-**lines 113-114** there is a repetition of what already reported at lines 107-108.

Sentence 107-108 removed

-**line 298** in the sentence "(…) *can also erode riverbanks and cause landslides, leading to further flooding*" the Author would probably mean the "breaking of the embankments/levees". This would be more correct than a general term "landslides".

Adjusted

-**line 323**: you could rewrite as "(….) *and stabilization of beaches and dunes*"

Adjusted

**-line 330-331**: "(….) *a continued decline in the extent of natural systems* (…)", this reduction in the extent can be very well expressed with the concept of "coastal squeeze".

Added

**-line 467**: I would delete citation "Romagnoli et al., 2022" from here (it is also mentioned just below as local-scale study).

Done

**-line 647**: in the caption of Fig. 11 it might be specified "(b) *current situation, with effects due to climate change and human activities*". Note that in the sketch c) the word "subsidence" should be corrected.

Added

**-line 693**: "(…) *such as due to subsidence*" would sound better.

Added

**-Figure 13**: this figure might be more extensively explained, it is very cryptic. SAR seems not mentioned in the text but only in the figure caption.

Thank you, we have improved the caption and text around the figure.

**-line 755**: a parenthesis is lacking at the end of the sentence "(*Figure 14*)".

Done

-**Figure 14**: correct in the caption "*upconing*", it is the scientific term for the process indicated in figure.

Done

-**line 774**: delete "other" from "*On the one hand*".

Done

---

## Referee Report (RR1)

I have reviewed the resubmitted version of the manuscript "*Sea level Rise in Europe: impacts and consequences*", providing an overview of the main type of impacts of SLR, with special focus on the European coasts. The manuscript has been partly re-organized; the Authors have accepted most suggestion provided by reviewers (although, for some unknown reason, in file "sp-2023-38-author_response-version1" I only see the rebuttal letter for reviewer 1, i.e. GLC, and not that for my previous review).

I indicate below a few comments and propose very minor further corrections. Some corrections derive from changes in the manuscript that the Authors made according to my previous comments on the original draft. However, some of these were probably not well expressed by me, or were not fully correct, and this generated some misunderstanding. I am sorry for that, and in this note I will update my early comments trying to better clarify them.

One of this, was my suggestion to add In **Table 1 -Summary of the methods for monitoring coastal erosion** a sentence on marine drones ("*Marine drones such autonomous surface vehicles (ASVs) are usefully applied for monitoring the nearshore in shallow water… etc etc*"). This sentence was inserted in the part of table dealing with "Drones", where Authors refer to the extensive collection of data covering larger areas along the coast with respect to ground-based surveys. Actually, this is not typically the case for Unmanned Vehicles in marine areas, that allow collecting information in autonomous manner, but at the spatial scale of "Field surveys". The use of marine drones has surely introduced various practical advantages in monitoring activites; however, it mostly fit the purpose "*for monitoring morphological changes in the short term and involves repeating measurements (….)*", indicated for "Field survey" techniques. So, the Authors might consider whether moving the reference to this application in other part of the table would appear more correct.

Some inconsistencies in the figures, or between text and figures, are still present:

-In introducing the common approach applied for the analysis, the term "Consequences" has been deleted in **figure 2** both from the sketch and caption according to my suggestion, and in section 4 the text now refers to the "SPR framework". However, some inconsistencies remain (such as in the caption of **figure 7**). Furthermore, despite the Authors state in their reply: "*Receptors and consequences are essentially the same thing and (…). As consequences are not referred to in the figures, we have as you suggested removed it.  Hence, we just refer to Source-Pathway-Receptor throughout*" and "*For homogeneity reasons we present all impacts using the Source, Pathway, Receptor framework*", in other parts of the manuscript than in section 4, the use of expression "Receptor and Consequences" (for instance, at lines 524 and 742) has been maintained in the text. The Authors still refer to the SPRC approach also in section "7 Conclusions", showing that this change has not been homogeneously adopted throughout the manuscript.

-**Figure 5**: I suggest deleting the original caption from below the figure; the figure explanation is already reported in your caption, while maintaining the original indication "Figure 1" is confusing.

-**Figure 6**: I still found not univocal the comparison between Figure 6 and the text describing this figure (lines 392-400). I see the indication you added "*for size in km$^2$ see Table 4 in Paprotny et al., 2019, for more details*" to clarify the estimates given in the text for the increase in flooded area. However, I do not find out in the figure other specific information you provide in the text: for instance, I do not recognize from the figure such a large value of flood extent area (over 4,500 km$^2$) for Norway (In Fig. 6 I see mostly green colors along the Norway coast) while you do not cite the coast of central Europe, where most of the blue color is present. Similarly, I do not find coherence

with colors of the figure and the indication "slightly below 2,000 km$^2$" for most part of Greece and Italy (apart the N Adriatic in correspondence of the Po Plain). Please check.

Minor comments and typo corrections:

-**line 70 and 429**: this reference should be "Bisaro et al 2024" and not "Galluccio et al 2024".
-**lines 71-73**: check the sentence: which is the subject of "(…) *can reduce the exposure*"? I suppose you wanted to refer to "*the establishment of coastal setback zones*" and not to "*Sano et al, 2011*…".
-**lines 78-79**: it might be indicated the reason why deltas are particularly vulnerable to SLR (such as due to the low altitude of coastal plain and natural subsidence).
-**line 157**: correct as "Coastal"
-**lines 177-179** and **193-195**: these two sentences basically repeat the same concept.
-**line 214**: check and correct "tides small tidal", it has no sense as it is.
-**lines 289-290** and **297-298**: again, these two sentences basically repeat the same concept.
-**lines 330-331** my original comment "line 323: you could rewrite as "(….) *and stabilization of beaches and dunes*")" was referred to the first part of the sentence that might become ("*building new or maintaining and improving existing flood defences, or application of artificial nourishment and stabilisation of beaches and dunes*"), and not to the second part of the sentence as it appears now, i.e. after " *and reduce flood risk along coasts*…". I probably was unclear. Check and correct.
-**line 469**: correct as "morphological"
-**line 497**: correct as "Receptor for" in the title of subsection 5.2.
-**line 540**: add "*(Italy")* after "*Emilia-Romagna coast*". Sorry, I did not specify this in my previous suggestion.
-**lines 635-636**: I remark the need for citing references here.
-**line 679 and 735**: I am afraid that the term "*saltine*" is not correct. I suggest using instead "*saline*", as you did at line 771 and 778).

I hope these further suggestions can be useful

Claudia Romagnoli, 12/06/2024

---

## Author Response (AR2)

Dear Claudia thanks for your rereview. In black the review in red our response

I have reviewed the resubmitted version of the manuscript "Sea level Rise in Europe: impacts and consequences", providing an overview of the main type of impacts of SLR, with special focus on the European coasts. The manuscript has been partly re-organized; the Authors have accepted most suggestions provided by reviewers (although, for some unknown reason, in file "sp-2023-38-author_response-version1" I only see the rebuttal letter for reviewer 1, i.e. GLC, and not that for my previous review).

Correct nearly all suggestion were accommodated

I indicate below a few comments and propose very minor further corrections. Some corrections derive from changes in the manuscript that the Authors made according to my previous comments on the original draft. However, some of these were probably not well expressed by me, or were not fully correct, and this generated some misunderstanding. I am sorry for that, and in this note I will update my early comments trying to better clarify them. One of this, was my suggestion to add In Table 1 -Summary of the methods for monitoring coastal erosion a sentence on marine drones ("Marine drones such autonomous surface vehicles (ASVs) are usefully applied for monitoring the nearshore in shallow water… etc etc"). This sentence was inserted in the part of table dealing with "Drones", where Authors refer to the extensive collection of data covering larger areas along the coast with respect to ground-based surveys. Actually, this is not typically the case for Unmanned Vehicles in marine areas, that allow collecting information in autonomous manner, but at the spatial scale of "Field surveys". The use of marine drones has surely introduced various practical advantages in monitoring activites; however, it mostly fit the purpose "for monitoring morphological changes in the short term and involves repeating measurements (….)", indicated for "Field survey" techniques. So, the Authors might consider whether moving the reference to this application in other part of the table would appear more correct.

Table 1 has been rephrased

Some inconsistencies in the figures, or between text and figures, are still present:

-In introducing the common approach applied for the analysis, the term "Consequences" has been deleted in figure 2 both from the sketch and caption according to my suggestion, and in section 4 the text now refers to the "SPR framework". However, some inconsistencies remain (such as in the caption of figure 7). Furthermore, despite the Authors state in their reply: "Receptors and consequences are essentially the same thing and (…). As consequences are not referred to in the figures, we have as you suggested removed it. Hence, we just refer to Source-Pathway-Receptor throughout" and "For homogeneity reasons we present all impacts using the Source, Pathway, Receptor framework", in other parts of the manuscript than in section 4, the use of expression "Receptor and Consequences" (for instance, at lines 524 and 742) has been maintained in the text. The Authors still refer to the SPRC approach also in section "7 Conclusions", showing that this change has not been homogeneously adopted throughout the manuscript.

The reviewer is correct we did not change it everywhere in the revised version. We went through the paper and corrected this.

-Figure 5: I suggest deleting the original caption from below the figure; the figure explanation is already reported in your caption, while maintaining the original indication "Figure 1" is confusing.

We now understand what you mean and followed your good suggestion and have taken the caption out of the figure.

-Figure 6: I still found not univocal the comparison between Figure 6 and the text describing this figure (lines 392-400). I see the indication you added "for size in km$_2$ see Table 4 in Paprotny et al., 2019, for more details" to clarify the estimates given in the text for the increase in flooded area. However, I do not find out in the figure other specific information you provide in the text: for instance, I do not recognize from the figure such a large value of flood extent area (over 4,500 km$_2$) for Norway (In Fig. 6 I see mostly green colors along the Norway coast) while you do not cite the coast of central Europe, where most of the blue color is present. Similarly, I do not find coherence with colors of the figure and the indication "slightly below 2,000 km$_2$" for most part of Greece and Italy (apart the N Adriatic in correspondence of the Po Plain). Please check.

We adjusted the explanation of the figure to clarify, the numbers are correct and refer to the 1971-2000 time slice and the 100-year return period.

Minor comments and typo corrections:

-line 70 and 429: this reference should be "Bisaro et al 2024" and not "Galluccio et al 2024".

corrected

-lines 71-73: check the sentence: which is the subject of "(…) can reduce the exposure"? I suppose you wanted to refer to "the establishment of coastal setback zones" and not to "Sano et al, 2011….".

Rephrased

-lines 78-79: it might be indicated the reason why deltas are particularly vulnerable to SLR (such as due to the low altitude of coastal plain and natural subsidence).

Added

-line 157: correct as "Coastal"

Done

-lines 177-179 and 193-195: these two sentences basically repeat the same concept.

Correct second sentence removed

-line 214: check and correct "Tides small Tidal", it has no sense as it is.

Corrected tides removed

-lines 289-290 and 297-298: again, these two sentences basically repeat the same concept.

Second one removed

-lines 330-331 my original comment "line 323: you could rewrite as "(….) and stabilization of beaches and dunes")" was referred to the first part of the sentence that might become ("building new or maintaining and improving existing flood defences, or application of arttificial nourishment and stabilisation of beaches and dunes"), and not to the second part of the sentence as it appears now ,i.e. aSer " and reduce flood risk along coasts…". I probably was unclear. Check and correct.

Adjusted

-line 469: correct as "morphological"

Corrected

-line 497: correct as "Receptor for" in thetiItle of subsection 5.2.

Corrected

-line 540: add "(Italy")) aSer "Emilia-Romagna coast". Sorry, I did not specify this in my previous suggetiIon.

Added

-lines 635-636: I remark the need for citing references here.

Done

-line 679 and 735: I am afraid that the term "saline" is not correct. I suggest using instead "saline",
as you did at line 771 and 778).

corrected

I hope these further suggestions can be useful

Thanks for your careful consideration of the manuscript

Claudia Romagnoli, 12/06/2024